# Shape evolution of bulk sediment in headwater streams: effects of rock type and particle size

Naoya O. Takahashi[1], Daisuke Ishimura[2, a], Keitaro Yamada[3], Ryoga J. Ohta[4,b], Yuki Arai[1], Yuki Yamane[1]

[1]Department of Earth Science, Tohoku University, Sendai, 980-0845, Japan

[2]Department of Geography, Tokyo Metropolitan University, Hachioji, Tokyo, 192–0397, Japan

[3]Research Centre for Palaeoclimatology, Ritsumeikan University, Kusatsu, Shiga, 525-8577, Japan

[4]The Institute of Science and Engineering, Chuo University, Tokyo, 112-8551, Japan

[a]present address: Graduate School of Science, Department of Earth Sciences, Chiba University

[b]present address: Faculty of Humanities, Niigata University,  Niigata 950-2181, Japan

*Correspondence to*: Naoya O. Takahashi (naoya.takahashi.c5@tohoku.ac.jp)

**Abstract.** Given the importance of sediments in fluvial morphodynamics, studying how sediment particle shapes change
during mass loss is important for understanding the morphology and change rates of fluvial landscapes. Particles of riverbed materials tend to become more rounded and circular downstream, but this trend can often be obscured because it results from processes that increase or decrease shape parameters to various degrees. Relative importance of the relevant processes, such as chipping, lateral sediment supply, and the production of finer particles during mass loss, may differ depending on rock type, lateral sediment supply, and particle size. This study evaluates the contributions of each process and reveals the factors
that determine the rates of change in shape parameters of riverbed material. We investigated changes in the roundness and circularity of basalt and shale particles in headwaters using the automated image analysis software Rgrains. Roundness is calculated using curvature of particle edges, while circularity is calculated using the entire perimeter of a particle. The observed pattern of downstream evolution of the roundness and circularity was clearly different between the upstream and downstream part of the studied area. Both shape parameters initially increased rapidly and remained nearly constant,
indicating that the dominant process affecting the particle shape changed during a few kilometers of transport. These punctuated shape changes result from the hillslope sediment supply and the addition of rock fragments produced by chipping and fragmentation, of which finer fragments were found to significantly alter the downstream evolution of the shape. This interpretation is supported by the fact that roundness of basalt particles slowly increased in the downstream part that lacks

supply of basalt particles from hillslopes. The rate of increase in the roundness and circularity of the particle shape depended on the rock type and grain size. The rates for the shale particles were higher than those for the basaltic particles. Grain size clearly affected the shape change rates of basalt particles but not of shale particles. We interpreted these differences between rock type and grain size to be associated with material strength, weathering mechanisms and speed, and total residence time in the channel. These findings demonstrate that image-based measurements of shape parameters in headwaters enables a detailed examination of the mechanism and rates of changes in particle shape.

## 1    Introduction

The size and shape of sediment particles dictates their entrainment threshold (Hattingh and Illenberger, 1995; Petit et al., 2015), transport mode (Sklar and Dietrich, 2004; Auel et al., 2017; Demiral et al., 2022), and flux (Vázquez-Tarrío et al., 2019; Cassel et al., 2021; Deal et al., 2023), which ultimately controls the channel hydraulic geometry and erosion rates (Hack, 1957; Sklar and Dietrich, 2004; Parker, 2007). Thus, understanding the evolution of fluvial landscapes requires knowledge of the downstream changes in particle size and shape and mechanisms of particle mass loss. Although many studies have focused on grain size (Paola et al., 1992; Kodama, 1994; Ferguson et al., 1996; Rice and Church, 1998; Attal and Lavé, 2006; 2009; Sklar et al., 2020), studies on shape are also crucial for identifying the causes of grain size changes downstream; this is because changes in size are usually accompanied by changes in shape. In addition, the particle shape reflects the transport history (Wentworth, 1923; Krumbein, 1941a; Benn and Ballantyne, 1994), depositional environment (Brook and Lukas, 2012; Lindsey et al., 2007; Ishimura and Yamada, 2019) and exhumation history (Quick et al., 2020), which enables the distinction of sediments of different origins and reconstruction of sediment provenance.

Among the many parameters to characterize two-dimensional particle shapes (Barrett, 1980; Blott and Pye, 2007), roundness and circularity are the most popular. Roundness is a measure of the corner curvature of a particle (Wentworth, 1919; Wadell, 1932), whereas circularity refers to the similarity of a particle outline to a perfect circle (Cox, 1927). Novák-Szabó et al. (2018) used three terms to describe attrition mechanisms under different energy conditions: frictional abrasion, chipping, and fragmentation, which were adopted in this study. Frictional abrasion occurs during rolling and sliding under low-energy conditions and causes flattening of the particle face (Novák-Szabó et al., 2018). Chipping removes a small piece of a particle, particularly from its corners and edges, thereby increasing both roundness and circularity (Domokos et al., 2014; Bodek and Jerolmack, 2021). Fragmentation occurs even under high-energy conditions; cracks formed by particle collisions develop throughout the particle, causing complete breakup of a particle (Bodek and Jerolmack, 2021). These three processes alter particle shape differently (Novák-Szabó et al., 2018).

The dominant attrition mechanism during transport and mass loss rate can change depending on factors, such as grain size, transport mode, material properties, and impact energy during collision (Kuenen, 1956; Zhang and Ghadiri, 2002; Attal and

Lavé, 2009; Novák-Szabó et al., 2018; Bodek and Jerolmack, 2021; Pfeiffer et al., 2022; Bray et al., 2024). In natural gravel-bed rivers, the bed materials consist of particles with different material properties and transport histories. Although the average roundness and circularity of amalgamated bed material generally increases downstream (Wentworth, 1923; Krumbein, 1941a; Pokhrel et al., 2024), lateral sediment supply from adjacent hillslopes and tributaries can disrupt this trend (Sneed and Folk, 1958), as observed for downstream changes in grain size (Rice, 1998; Rice and Church, 1998). Irregular changes in shape parameters owing to lateral sediment supply preclude the estimation of the rates of change in shape parameters with transport distance (Russel and Taylor, 1937; Sneed and Folk, 1958; Mills, 1979).

Given the multitude of processes operating in natural rivers, the dominant processes altering the particle shape can change during transport (e.g., Knighton, 1982). This makes the evaluation of the contributions of each process and identification of the causes of changes in particle shape challenging. Although focusing on a short section of a river may help reduce the number of contributing factors, it requires a very accurate measurement of shape parameters because the absolute changes in roundness and circularity are on the orders of $10^{-2}$–$10^{-1}$ (Litwin Miller et al., 2014; Novák-Szabó et al., 2018) Thus, a study that can link field investigations with laboratory and theoretical studies focusing on the effects of individual processes is required.

One solution is to study the shape evolution of headwaters using image analysis. Both laboratory and field studies have shown that the rate of change in some shape parameters decreases with the total transport distance or as the particle loses its mass (Sternberg, 1875; Wentworth, 1923; Krumbein, 1941a; Domokos et al., 2014; Litwin Miller et al., 2014; Novák-Szabó et al., 2018; Bodek and Jerolmack, 2021; Bray et al., 2024). For instance, Bray et al. (2024) investigated the mass loss rates of bulk sediment using a tumbling mill and found that the rates decreased by up to one order of magnitude when the bulk mass was reduced to approximately 90% of its initial value. Thus, headwaters are the best locations for observing significant changes in particle shape over short transport distances.

The advent of 2D and 3D image analysis tools has enabled the efficient and reproducible measurement of particle shapes (Roussillon et al., 2009; Zheng and Hryciw, 2015; Fehér et al., 2023; Cattapan et al., 2024; Tripathi et al., 2025). In particular, 2D image analysis can improve the accuracy of shape measurements by dramatically increasing the number and size of samples. Litwin Miller et al. (2014) conducted manual and image-based measurements of particle shape in headwaters at over 60 sites along a 10 km reach. Despite some scatter in the measured shape parameters, they successfully identified changes in circularity of <0.1 and those in increasing rates of circularity. A large sample size also enables the extraction of information from the distribution of shape parameters. Although the distribution of shape parameters potentially provides insights into the origin of the sediments (Sneed and Folk, 1958; Ishimura and Yamada, 2019), tremendous effort is required to construct smooth histograms, which is impractical when conducting conventional shape measurements (e.g., Krumbein, 1941b; Powers, 1953). Constructing smooth histograms is particularly important for studying the spatial changes in shape distributions (Ishimura and Yamada, 2019). Ishimura and Yamada (2019) measured the roundness of $>10^4$ particles in tsunami deposits by using an image analysis algorithm developed by Zheng and Hryciw (2015). They showed that the sediments of different beaches and rivers can be distinguished using the distribution of

roundness. They also revealed the inundation heights of paleotsunamis using the ratio of marine and fluvial sediments estimated from the roundness distributions. Therefore, image-based measurement of the particle shape would benefit the evaluation of the relative contributions of relevant processes to the downstream changes in the shape parameters.

Motivated by these ideas, we investigated the downstream changes in the roundness and circularity of headwaters using image analysis. We aimed to reveal (1) what processes cause a downstream change in roundness (Wadell, 1932) and circularity (Cox, 1927) and (2) the effects of rock type and particle size on the rates of change in roundness and circularity. We collected samples of four size classes from the main stream, tributaries, and talus deposits to study the evolution of shape from the beginning of transport. We also performed a manual crushing experiment to estimate the shapes of the particles produced by chipping and fragmentation, which was used to evaluate how these processes affect the distributions of the shape parameters. All samples were separated into two rock types, shale and basalt, and analyzed using the image analysis software Rgrains (Yamada, 2024). We demonstrated that the rate of increase in roundness and circularity dropped significantly after the initial 2 km of transport. While the spatial pattern of change was common to both rock types and grain size classes, their rates of change differed. We evaluated the contributions of the processes that affect the distribution of shape parameters to the observed changes in particle shape and discussed the causes of the observed shape evolution and dependence of the particle shape on the rock type and grain size.

## 2 Geologic background

We studied a small river, the Mosawa, located on Tsugaru Mountain in northeastern Japan (Fig. 1). Mosawa is a perennial river and receives 1400 mm $y^{-1}$ of rain on average (Japan Meteorological Agency, 2025). We focused on a 6 km-long reach from the channel head, which drains an area of 4.2 $km^2$ (Figs. 1a and 1b). Channels in the studied area are single-threaded and less than 10 meters wide (Figs. 1c and 1d). Miocene basalts and dolerites dominate the upstream part of the studied catchment, and Miocene shale dominates the downstream part (Tsushima and Uemura, 1959; Uemura et al., 1959). The basaltic rocks exposed along streams exhibit various degrees of fracturing and weathering (Tsushima and Uemura, 1959; Uemura et al., 1959). Shale is prone to slaking. The fresh part of the shale is dark gray and turns white when weathered. The exposed bedrock of shale and shale particles found in the bed are mostly white and much more fragile than the basaltic particles. The channel slopes tend to be steeper on basaltic rocks than on shales. A check dam is located in the downstream area (Fig. 1a). We observed that coarse grains were mostly trapped in the upstream half of the dammed section, and only fine grains less than several millimeters in diameter were present near the dam wall. Thus, we mainly focused on the reaches upstream of the dam.

Four tributaries with catchment areas >0.1 $km^2$ supply gravels to the main stream in the studied reach. The catchment area of the most upstream tributary is comparable to that of the main stream at its confluence. The catchment areas of the other three tributaries are 10–34% of that of the main stream at the confluences. Shale particles are supplied from all four tributaries,

whereas basaltic particles are supplied only from two tributaries. These differences in the relative catchment area and lithology suggest that the impact of tributary supply on the evolution of particle shape in the main stream varies among tributaries (Rice, 1998).

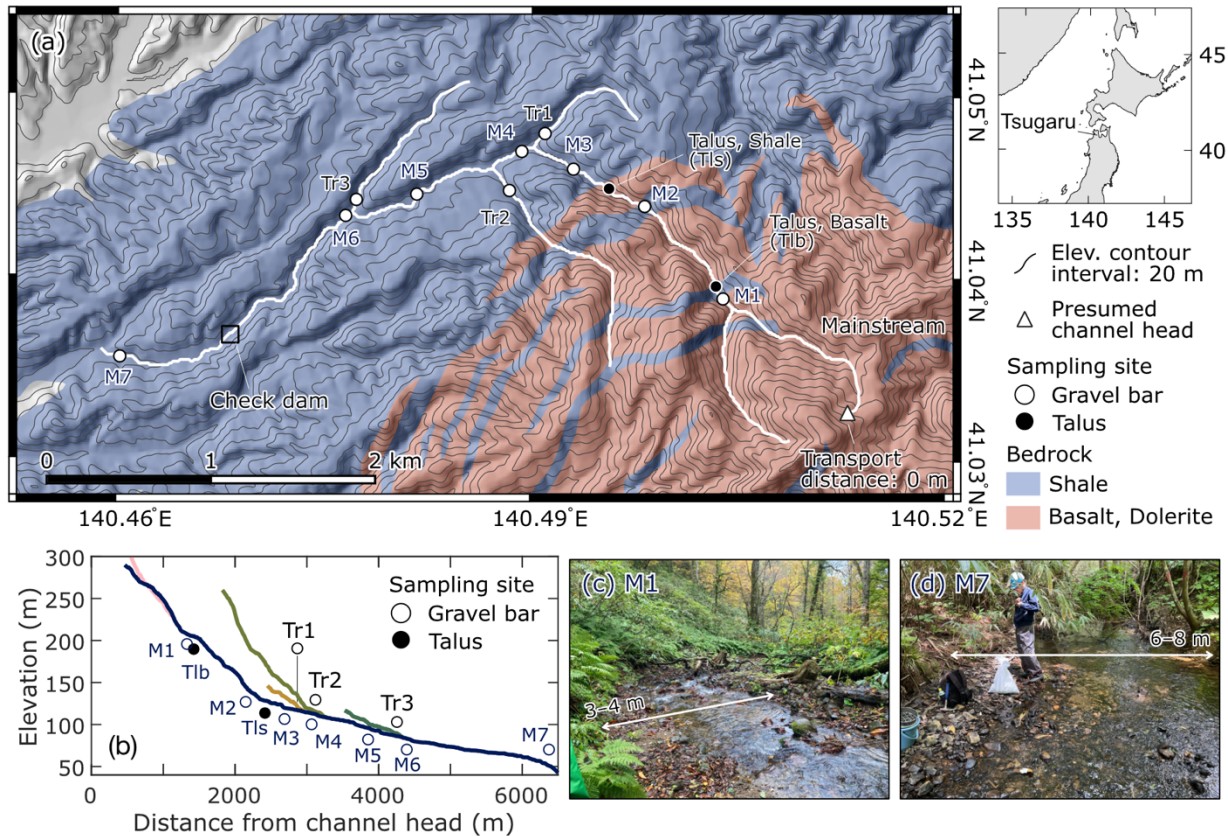

Figure 1: (a) Geologic map of the study area. Geologic units are after Tsushima and Uemura (1959) and Uemura et al. (1959). (b) Longitudinal profiles of the main stream and tributaries. Triangles indicate the sampling site locations. (c) (d) The most upstream and downstream sampling site, respectively.

## 3 Materials and Methods

### 3.1 Definition of shape parameters and downstream evolution model

In this study, we used the roundness and circularity of the particles. We used Waddell's roundness ($R$), a ratio of average radius of curvature of the corners ($r_c$) and radius of the maximum inscribed circle ($r$) (Wadell, 1932).

$$R = \frac{r_c}{r} \tag{1}$$

Following Wadell (1932), we defined a corner used to calculate $R$ as a part of the particle outline that has a radius of curvature equal to or less than the radius of curvature of the maximum inscribed circle of the particle. Circularity is the degree to which a two-dimensional outline of a particle is close to that of a circle (Blott and Pye, 2007). Cox (1927) introduced a proxy for roundness, which has also been termed the isoperimetric ratio ($IR$) in recent studies on particle shapes (Litwin Miller et al., 2014; Quick et al., 2020; Pokhrel et al., 2024)

$$IR = \frac{4\pi A}{P^2}, \tag{2}$$

where $A$ is the projected area of a particle and $P$ is its perimeter. $IR$ represents how close a particle shape is to a circle and is 1 for a perfect circle. Blott and Pye (2007) argued that circularity is a more accurate term for Cox's roundness ($IR$), because $IR$ is a measure of the overall particle shape rather than individual corners. We follow Blott and Pye (2007) and use $IR$ as a proxy for circularity because roundness and circularity can evolve downstream in different manners (Wentworth, 1923; Russel and Taylor, 1937; Krumbein, 1941a; Pettijohn and Lundahl, 1943; Sneed and Folk, 1958).

Particle roundness and circularity tend to increase nonlinearly with the transported distance (e.g., Wentworth, 1923; Krumbein, 1941a; McPherson, 1971; Mills, 1979; Novák-Szabó et al., 2018; Pokhrel et al., 2024). Krumbein (1941a) proposed a formula for the downstream evolution of roundness with transport distance $x$ (km).

$$R = R_{lim} - (R_{lim} - R_0)e^{-\lambda_R x}, \tag{3}$$

where $R_{lim}$ is the limiting roundness, $R_0$ is the initial roundness before transportation, and $\lambda_R$ is a coefficient that controls the rate of change in $R$. The downstream evolution of $IR$ follows the same trend as in Eq. (3) (Krumbein, 1941a).

$$IR = IR_{lim} - (IR_{lim} - IR_0)e^{-\lambda_{IR} x}, \tag{4}$$

where $IR_{lim}$ is the limiting $IR$, $IR_0$ is the initial $IR$ before transportation, and $\lambda_{IR}$ is a coefficient. Although Eqs. (3) and (4) do not perfectly match the observed changes in $R$ and $IR$ especially in the downstream half of the studied reach, we used Eqs. (3) and (4) because they were useful to show single values of $\lambda_R$ and $\lambda_{IR}$ cannot fully explain the observed changes in $R$ and $IR$. Assuming $R_{lim}$ and $IR_{lim}$ are 1, Eqs. (3) and (4) can be rearranged as follows:

$$\frac{(1-R)}{(1-R_0)} = e^{-\lambda_R x}, \tag{5}$$

$$\frac{(1-IR)}{(1-IR_0)} = e^{-\lambda_{IR} x}. \tag{6}$$

The left-hand side of these equations is the ratio of the difference between the limiting shape parameters and those at distances $x$ and 0. Changes in $\lambda_R$ and $\lambda_{IR}$ were readily observed in semi-log plots (Krumbein, 1941a).

## 3.2 Sampling

We collected particles from gravel bars at seven sites in the trunk stream (M1 to M7), three sites in the tributaries (Tr1 to Tr3), and two taluses (Tlb and Tls, Fig. 1b). All sampling sites, except for one, were located upstream of the check dam. Talus samples were collected to study the initial distribution of particle shapes when supplied from the hillslopes. Tributary samples were collected to study impacts of sediment supply from tributaries on the shape evolution in the mainstream. We collected particles sized 2–4, 4–8, 8–16, and 16–32 mm using square mesh sieves. Samples downstream of the dam (M7) were collected to study downstream changes in particle shape over longer distances. However, particle shape downstream of the dam may be altered while transported over the dam sediment. Thus, only particles with the finest size range of 2–4 mm were collected. This is because we only found particles of several millimeters immediately downstream of the dam wall, and the rock type composition of the 2–4 mm samples did not change across the dam (basaltic particles constituted 71% and 66– 82% of the samples taken downstream and upstream of the dam, respectively). The samples were then dried and separated into shale and basaltic particles based on their color, texture, and hardness. For simplicity, we did not differentiate basalt from dolerite, classifying both as basalt. The number of particles analyzed was greater for smaller particles and basalt (Table 1). The average number of particles was 3157 and 4938 for the 2–4 mm particles of shale and basalt, respectively, and 253 and 472 for the 16–32 mm particles of shale and basalt, respectively. In total, we analyzed 152932 particles.

**Table 1: The number of particles measured in this study. Samples with labels M, Tr, and Tl are those taken from the main stream, tributaries, and talus deposit, respectively.**

| Sample | 2–4 mm Shale | Basalt | 4–8 mm Shale | Basalt | 8–16 mm Shale | Basalt | 16–32 mm Shale | Basalt |
|--------|------|--------|------|--------|------|--------|------|--------|
| M1 | 1915 | 4283 | 946 | 4002 | 688 | 1396 | 185 | 380 |
| M2 | 2097 | 4000 | 1218 | 3822 | 717 | 1528 | 165 | 583 |
| M3 | 1750 | 7187 | 801 | 2536 | 694 | 1323 | 144 | 519 |
| M4 | 1229 | 5224 | 1166 | 3375 | 891 | 1386 | 107 | 486 |
| M5 | 1301 | 3763 | 1246 | 2928 | 658 | 1394 | 220 | 436 |
| M6 | 1304 | 6122 | 1465 | 3890 | 719 | 1170 | 233 | 530 |
| M7 | 2568 | 6426 | n/a | n/a | n/a | n/a | n/a | n/a |
| | | | | | | | | |
| Tr1 | 6078 | n/a | 3832 | n/a | 1321 | n/a | 309 | n/a |
| Tr2 | 2904 | 3598 | 1245 | 3040 | 610 | 1305 | 248 | 480 |
| Tr3 | 6588 | n/a | 2086 | n/a | 866 | n/a | 330 | n/a |
| | | | | | | | | |
| Tls | 6996 | n/a | 5057 | n/a | 2493 | n/a | 593 | n/a |
| Tlb | n/a | 3841 | n/a | 4248 | n/a | 1386 | n/a | 362 |

### 3.3 Image analysis

The image analysis procedure involved photographing, pre-processing images, and analyzing particle shapes using Rgrains v4.11. (Ishimura and Yamada, 2019; Yamada, 2024). Workflow of the analysis is summarized in Figure 2. Rgrains is an image analysis software built using an algorithm developed by Zheng and Hryciw (2015). The major difference between the scripts provided by Zheng and Hryciw (2015) and Rgrains is that Rgrains can be run through GUI and without a MATLAB license. The core algorithm of Rgrains to measure particle shape is the same as the one developed by Zheng and Hryciw (2015). Readers are referred to Zheng and Hryciw (2015) for the details of the calculation and to Yamada (2024) for how to use Rgrains.

190

## (a) Taking photograph

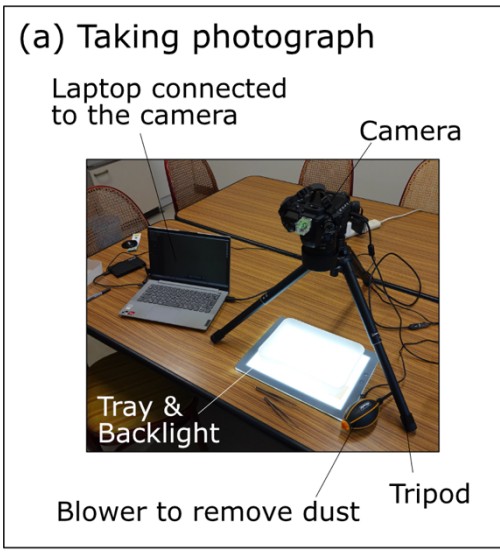

Laptop connected to the camera
Camera
Tray & Backlight
Blower to remove dust
Tripod

## (b) Image preprocessing

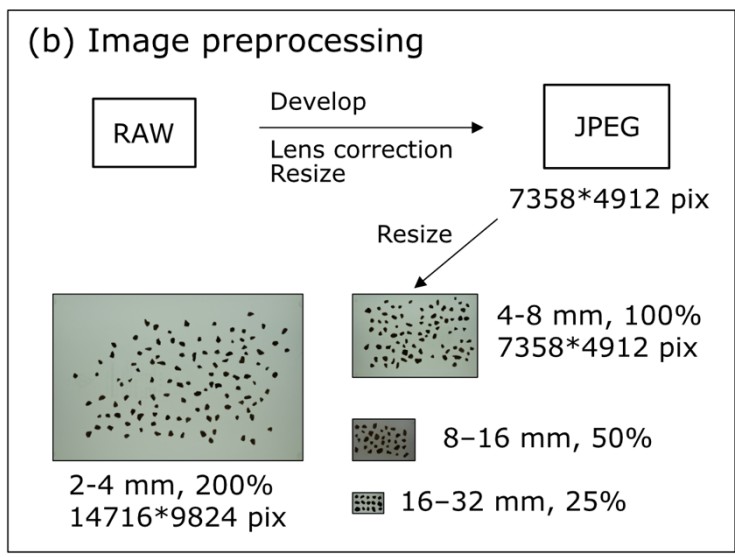

RAW →(Develop / Lens correction / Resize)→ JPEG

7358*4912 pix

Resize

4-8 mm, 100%
7358*4912 pix

2-4 mm, 200%
14716*9824 pix

8–16 mm, 50%

16–32 mm, 25%

## (c) Image analysis using Rgrains

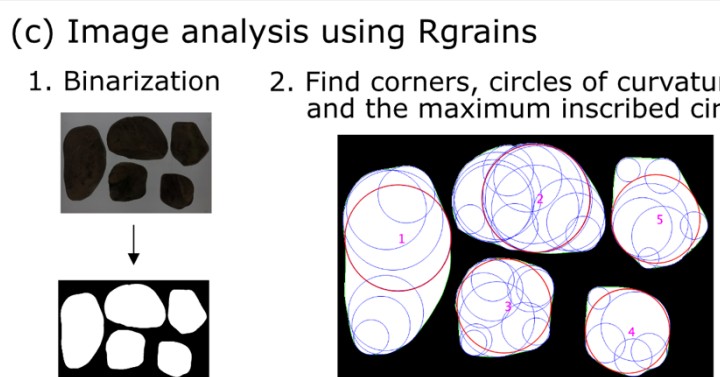

1. Binarization
2. Find corners, circles of curvature, and the maximum inscribed circle
3. Compute shape parameters and export as a csv file
   - Wadell's roundness
   - Isoperimetric ratio
   - Major axis length
   - Minor axis length
   …etc

## (d) Check output images and remove wrong data

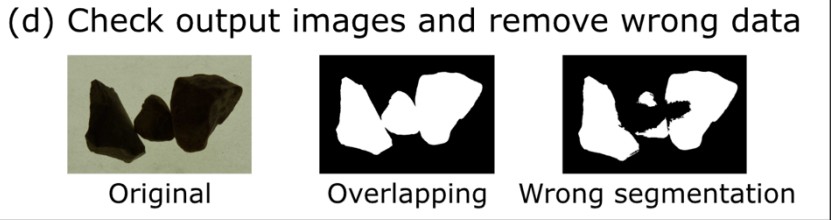

Original     Overlapping     Wrong segmentation

195

**Figure 2: A workflow of the image analysis. (a) A camera setup to take particle images. (b) Resizing images according to the particle size classes. (c) Input images are first binarized to separate particles and the background. The image is then analyzed based on an algorithm developed by Zheng and Hryciw (2015). (d) Erroneous results such as particles overlapping with others and those wrongly segmented are manually removed.**

200

Before taking photographs, we aligned the particles on the tray while avoiding overlap. Ideally, particles should be aligned on a tray based on unified criteria, such as with their maximum projected areas parallel to the tray. However, in practice, it is

not possible to find the maximum projection plane of each particle manually, and many natural particles cannot stand still with their maximum projected plane upward. Thus, we used a particle facet that a particle can stand stably and with the maximum projection areas among all stable facets. The tray was placed on a light pad, and a backlit photo was taken from above using a camera mounted on a tripod (Fig. 2a). A camera and lens we used was Nikon D750 and Micro Nikkor 60 mm, f/2.8G ED, respectively. We remotely controlled the camera using a free application digiCamControl to adjust the focus and check images instantly. Backlit photos are used to increase the contrast between particles and the background, which is favorable for image segmentation (Cassel et al., 2018). Because the results of the image analysis depend on the image resolution (Zheng and Hryciw, 2015), we adjusted the camera height such that the image resolutions for the samples were similar.

During preprocessing, we first converted the raw images to JPEG images with dimensions of 7358 and 4912 pixels using Adobe Photoshop and Affinity Photo (Fig. 2b). After conversion, the image resolutions for the 74 samples ranged between 335 and 348 pix cm$^{-1}$, with the exception of eight samples with resolutions of 312–319 pix cm$^{-1}$. We confirmed these differences were unlikely to affect the results, which would be further explained in the next paragraph. Adobe Photoshop and Affinity Photo can be replaced with any image processing software. Results of image analysis depend on the number of pixels per diameter of a circumscribed circle of a particle (PCD: Zheng and Hryciw, 2015). To minimize the dependency of the image analysis on the PCD, we adjusted the PCD values for the 16–32, 8–16, and 2–4 mm particles to those for 4–8 mm particles (Fig. 2b). Images of 16–32 and 8–16 mm particles were downscaled to 25% and 50% of the original resolutions, respectively, using Adobe Photoshop and Affinity Photo. Images of 2–4 mm particles were upscaled to 200% of the original resolution by bicubic interpolation using Rgrains. Rgrains uses a MATLAB function "imresize" to double the image resolution. After PCD adjustment, particles with a PCD <200 were omitted because their outlines could not be properly identified (Zheng and Hryciw, 2015). Although the particles analyzed using lower-resolution images had slightly lower PCD values than those analyzed using higher-resolution images, such differences were unlikely to affect the results because the values of roundness and circularity did not change with the PCD as long as the PCD exceeded 200 (Zheng and Hryciw, 2015).

The shape parameters were then calculated using the Rgrains. We first binarized jpeg images using adaptive (Bradley and Roth, 2007) or Otsu (Otsu, 1979) methods and separated the particle and background regions (Fig. 2c). The extracted particle regions greater than the threshold size were analyzed. The threshold size was adjusted to remove rock fragments that were significantly smaller than the sieve openings. The scales of the images were adjusted according to the original scale of the image and the degree of up- and downscaling. For the other parameters, we used the default settings for Rgrains following the suggestions of Ishimura and Yamada (2019). Rgrains then found corners and computed circles of curvature of each corner and the maximum inscribed circle using an algorithm developed by Zheng and Hryciw (2015). A summary of this process can be found on a GitHub page (Yamada, 2024). Finally, Rgrains calculated $R$ and $IR$ of each particle. After the calculation, we checked binarized images and removed results of overlapping particles and those improperly segmented (Fig. 2d).

### 3.4 Downstream evolution and changing rates of shape parameters

To estimate the rates of changes in $R$ and $IR$ ($\lambda_R$ and $\lambda_{IR}$), we fitted Eqs. (3) and (4) to the median $R$ and $IR$ values using the least squares method. We used only median values because distributions of $R$ and $IR$ were skewed, and changes in 14th and 86th percentiles values of $R$ and $IR$ were essentially similar to a change in the median $R$ and $IR$ values (Takahashi et al., 2025). We defined the location of the channel head as a point at the valley bottom where 1 m elevation contours bend at an acute angle (Hattanji et al., 2021). Distance from the channel head was used as the transport distance $x$ in Eqs. (3) and (4). The $R$ and $IR$ values of the talus samples were used as the initial $R$ and $IR$ values at the channel head ($x = 0$). Thus, positions of the shale and basalt talus samples plotted in subsequent figures are different from the actual sampling positions. Similarly, results of samples taken from the tributaries are plotted at their confluences with the mainstream. Thus, their horizontal coordinates are different from distance from channel heads of each tributaries. Although shale units were not mapped in the most upstream section in the available geological map (distance <1000 m in Fig. 1b), the Wolman count results showed that shale gravel constituted 12–14% of the bed materials in that section (Takahashi, 2025). Thus, we assumed the same channel head positions ($x = 0$) for shale and basalt particles.

To identify the cause of the changes in particle shape, we compared the histograms of $R$ and $IR$ between adjacent sampling sites. The shape of the histogram depends on the choice of bin positions, implying that the calculated changes in relative frequency could be an artifact. To minimize these effects, we used a moving histogram to calculate the changes in relative frequency downstream. We first set the width and the minimum position of the bin to 0.05 and calculated the downstream changes in the relative frequency of a given value of the shape parameters. We then shifted the bin position by one-fiftieth of the bin width and calculated the difference in relative frequency. This process was repeated 49 times. The centers of each bin and their differences in relative frequency were plotted together to form a composite histogram showing downstream changes in particle shape distribution between adjacent sampling sites.

Given the short distance of the studied reach, quantifying uncertainties in the measured shape parameters are important to interpret the observed evolution of particle shape. To estimate the difference between the median $R$ and $IR$ values of the samples and the populations, we examined changes in sample median $R$ and $IR$ values with sample size. We used 2–4 mm samples with the maximum and minimum standard deviation of $R$ and $IR$ to estimate the uncertainty range of median $R$ and $IR$ values. The number of 2–4 mm particles are much greater than those of other size classes (Table 1). Thus, we assumed median $R$ and $IR$ values of the 2–4 mm samples (hereafter, true median values) were the closest to those of the population among our samples. For $R$, we used shale particles of M7 and basalt particles of Tlb. For $IR$, we used shale particles of M7 and basalt particles of M5. For each sample, we extracted the specified number of shape parameters while allowing duplication and calculated the median value. The number of extracted data (i.e. sample size) ranged between 10–3000. For each sample size, we obtained 5000 median values and calculated their 2.5th and 97.5th percentile values. We then subtracted

the true median value from the 2.5th and 97.5th percentile values, which we consider the maximum deviation of the sample median value from the true median value.

## 3.5 Manual crushing experiment

To evaluate the impact of chipping and fragmentation on the shape evolution of bulk sediment, we conducted a manual crushing experiment. The objectives of this experiment were to reveal the shape distribution of fine rock fragments produced by chipping and fragmentation and the effects of the initial shape distributions on those of the rock fragments. The shape distribution obtained was assumed to be comparable to those produced by natural rivers. This assumption is based on the findings of Domokos et al. (2015) that the distributions of the shape and mass of rock fragments are similar regardless of how the fragments are produced. We used 8–16 mm shale particles taken from the talus and the second-most downstream sites, presumably the least and most rounded and circular samples, respectively. We chose 8–16 mm particles because coarser gravel can produce a greater number of rock fragments, and we could not bring 16–32 mm particles to the laboratory for logistical reasons. When crushing the particles, we smashed the hammer gently and strongly enough to break the particles with a single smash to minimize changes in the particle shape such as flattening of edges. We did not use basalt particles because they are too hard to crush easily. The produced rock fragments were sieved and analyzed in the same manner as the other samples. We focused on rock fragments sized 2–4 mm and compared their shape distributions with those before crushing. We also compared crushed particles with non-crushed 2–4 mm talus particles to determine whether the hillslope sediment supply and production of rock fragments due to chipping and fragmentation had similar impacts on the shape distribution in the main stream.

To examine the difference in shape parameters between the two crushed 2–4 mm samples and the original talus sample of the same size, we performed a Wilcoxon rank sum test. The null hypothesis was that the two samples would have the same median values. The sample sizes of the crushed talus, crushed fluvial, and original 2–4 mm talus samples were 11558, 3266, and 6996, respectively, suggesting that the p-values of the test can be very small even if the two samples have almost identical median $R$ or $IR$ values. To determine whether the p-values changed with sample size, we randomly extracted the specified amount of data from each sample while allowing duplication and performed a Wilcoxon rank sum test. The sample size varied from 100 to 50000. For each sample size, we repeated these steps 50 times and obtained 50 p-values for three pairs of samples: crushed talus–crushed fluvial, crushed talus–original talus, and crushed fluvial–original talus.

# 4 Results

## 4.1 Shape evolution along the main stream

### 4.1.1 Roundness

The major differences between shale and basalt are that shale particles are more rounded and finer basalt particles have lower $R$ values, whereas the $R$ of shale particles are similar across the four grain size classes (Figs. 3a and 3b). Except for these two differences, the downstream evolutions of $R$ for shale and basalt particles were essentially similar. The initial roundness of the talus deposits was significantly lower than that of the fluvial samples. Downstream of the channel head, the roundness rapidly increased during the initial 2 km of transport. The median $R$ then remained constant or decreased in the middle

section between 2060 to 3780 m (M2 to M5). In the further downstream section (M4 to M7), the median $R$ of basalt samples increased. The magnitude of this change was much smaller than that in the most upstream section (< 2060 m, channel head to M2).

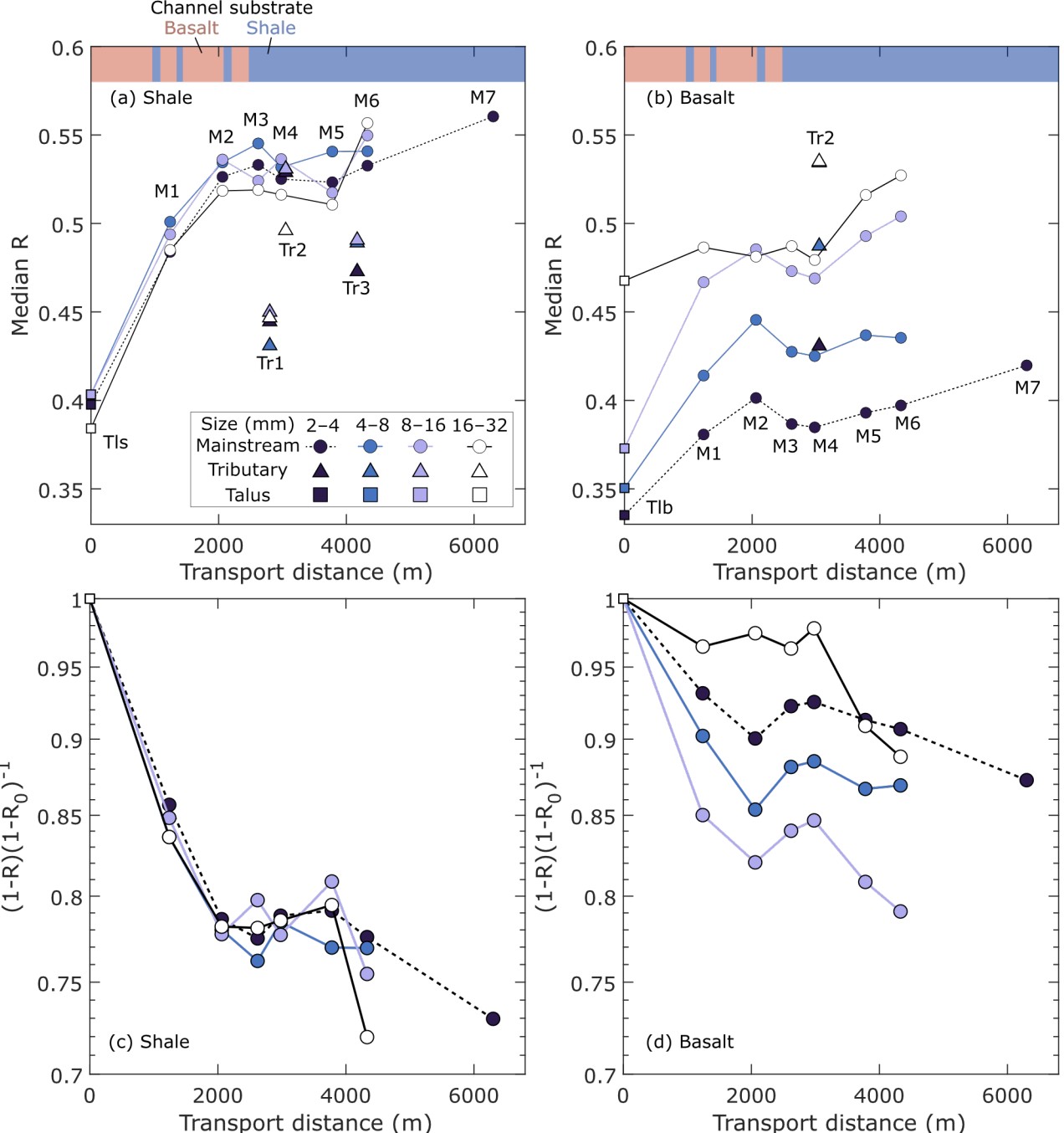

**Figure 3: (a, b) Downstream evolution of roundness. (c, d) The ratio of the difference between the limiting roundness**

**and measured roundness at each sampling site ($R$) and channel head ($R_0$). Tributary data are plotted at the**

**confluences with the main stream. Talus data are plotted at the presumed channel heads, not at the actual sampling locations. The vertical axis in (c) and (d) is in a logarithmic scale.**

Plotting the left-hand side of Eq. (5) against transport distance revealed changes in the coefficients ($\lambda_R$) within the studied section (Figs. 3c and 3d; Table 2). The rounding coefficient $\lambda_R$ estimated using all samples in the main stream was larger for shale particles than for basalt (Table 2). The p-values for $\lambda_R$ were 0.007–0.038, indicating the observed downstream evolution of $R$ is consistent with the general trend (Eq. 3). We also fitted Eq. (3) to data from upstream (< 2060 m, channel head to M2) and downstream section (> 2620 m, M3 to M7). Although the limited number of samples makes it difficult to estimate $\lambda_R$ and its p-value correctly, the values of $\lambda_R$ were probably much larger in the upstream section than in the downstream section (Table 2). In the upstream section where the $R$ values change rapidly, the $\lambda_R$ values for shale particles did not depend on particle size, whereas those for basaltic particles increased with particle size, except for those sized 16–32 mm (Table 2). In the downstream section, p-values for shale were 0.07–0.97 and much larger than in the upstream section, which suggests the median $R$ values for shale particle may not change significantly in the downstream section. The p-values for basalt in the downstream section were much smaller than those for shale in the downstream section. In particular, the increase in $R$ for 2–4 mm particles of basalt is probably significant (p=0.003).

**Table 2: Rates of change in roundness.**

| | All | | Channel head to M2 (< 2060 m) | | M3 to M7 (2620–6300 m) | |
|---|---|---|---|---|---|---|
| Extent (distance) | | | | | | |
| Grain size (mm) | $\lambda_R$ (km-1) | p | $\lambda_R$ (km-1) | p | $\lambda_R$ (km-1) | p |
| Shale | | | | | | |
| 2–4 | 0.04 | 0.008 | 0.12 | 0.030 | 0.02 | 0.070 |
| 4–8 | 0.06 | 0.017 | 0.12 | 0.084 | 0.00 | 0.973 |
| 8–16 | 0.05 | 0.017 | 0.12 | 0.037 | 0.02 | 0.529 |
| 16–32 | 0.06 | 0.008 | 0.12 | 0.088 | 0.04 | 0.330 |
| | | | | | | |
| Basalt | | | | | | |
| 2–4 | 0.02 | 0.007 | 0.05 | 0.054 | 0.02 | 0.003 |
| 4–8 | 0.03 | 0.038 | 0.08 | 0.035 | 0.01 | 0.144 |
| 8–16 | 0.05 | 0.013 | 0.10 | 0.151 | 0.04 | 0.042 |
| 16–32 | 0.02 | 0.017 | 0.01 | 0.433 | 0.06 | 0.057 |

Figure 4 shows the changes in the roundness distribution between adjacent sites for 2–4 mm basalt particles. All other

histograms of the individual sites and differential histograms are presented by Takahashi et al. (2025). The shapes of these

differential histograms differed between the upstream, middle, and downstream sections of the study area.

In the upstream section, where $R$ increased rapidly (>2060 m, channel head to M2), the differential histogram contained

single positive and negative peaks at higher and lower roundness, respectively (Fig. 4a). Figure 4b shows the changes in the

roundness distribution when the net decrease in the median $R$ is the greatest. When the median $R$ values were almost

constant, the differential histograms often consisted of multiple ridges and troughs (Fig. 4c). The complex shapes of the

differential histograms were partly due to the rough shapes of the original histograms. However, these complex changes

were found even for 2–4 mm particles, whose minimum sample size exceeded 1000, implying that multiple processes were

involved in the observed changes in $R$. When the median $R$ increased in the most downstream section (Fig. 4d), the

differential histogram is characterized by less prominent peaks than in the upstream section (Fig. 4a) and a roughly constant

increase for $R$ between 0.55 and 0.74.

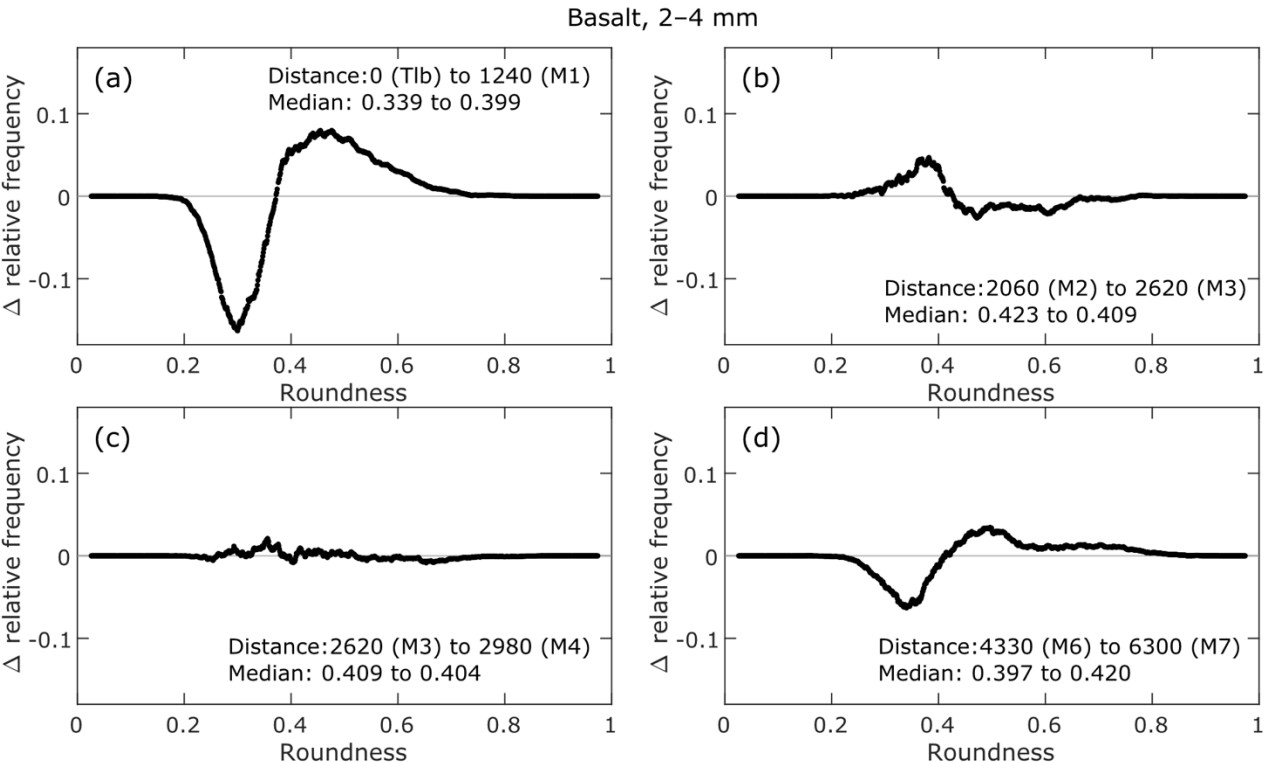

**Figure 4: Changes in the roundness ($R$) distribution between adjacent sites. The distance between the two sampling sites and their median $R$ values are provided in the lower-right of each panel.**


The three tributaries supplied particles with different roundness to the main stream (Fig. 3). The shale particles were less rounded in the three tributaries than in the main stream, potentially contributing to the marginal increase in $R$ in the middle part of the study area. However, the changes in the median $R$ values were not always consistent with the supply of less-rounded particles from the tributaries. For instance, the transition from initial rapid rounding to punctuated rounding occurs

upstream of any tributary junction. In addition, the median $R$ values increased significantly between 3780 and 4330 m (M5 to M6), where particles with lower $R$ were supplied from the tributary (Tr3) to the main stream. These results indicate that tributary supply alone cannot explain the changes in $R$ values for shale in the main stream. In contrast, an increase in $R$ of the basalt particles occurred concurrently with the supply of rounded particles at the tributary confluence at 3220 m (Tr2, Fig. 3b). However, the rates of change in $R$ across this confluence were not significantly higher than those in the further

downstream section, which lacked tributary inputs of basalt particles. Thus, tributary supply alone cannot explain the changes in $R$ values for basalt in the main stream.

### 4.1.2 *IR*

The downstream changes in *IR* was less pronounced compared to the changes in $R$ (Fig. 5). The absolute difference in *IR*

between the talus and fluvial samples in the main stream was smaller than that in $R$ (Figs. 3 and 5). The *IR* values of both rock types increased rapidly during the first 2 km of transport and stopped increasing thereafter. Unlike roundness of basalt, the median *IR* did not evidently increase in the most downstream section, as suggested by higher p-values for $\lambda_{IR}$ than those for $\lambda_R$ (Table 3).

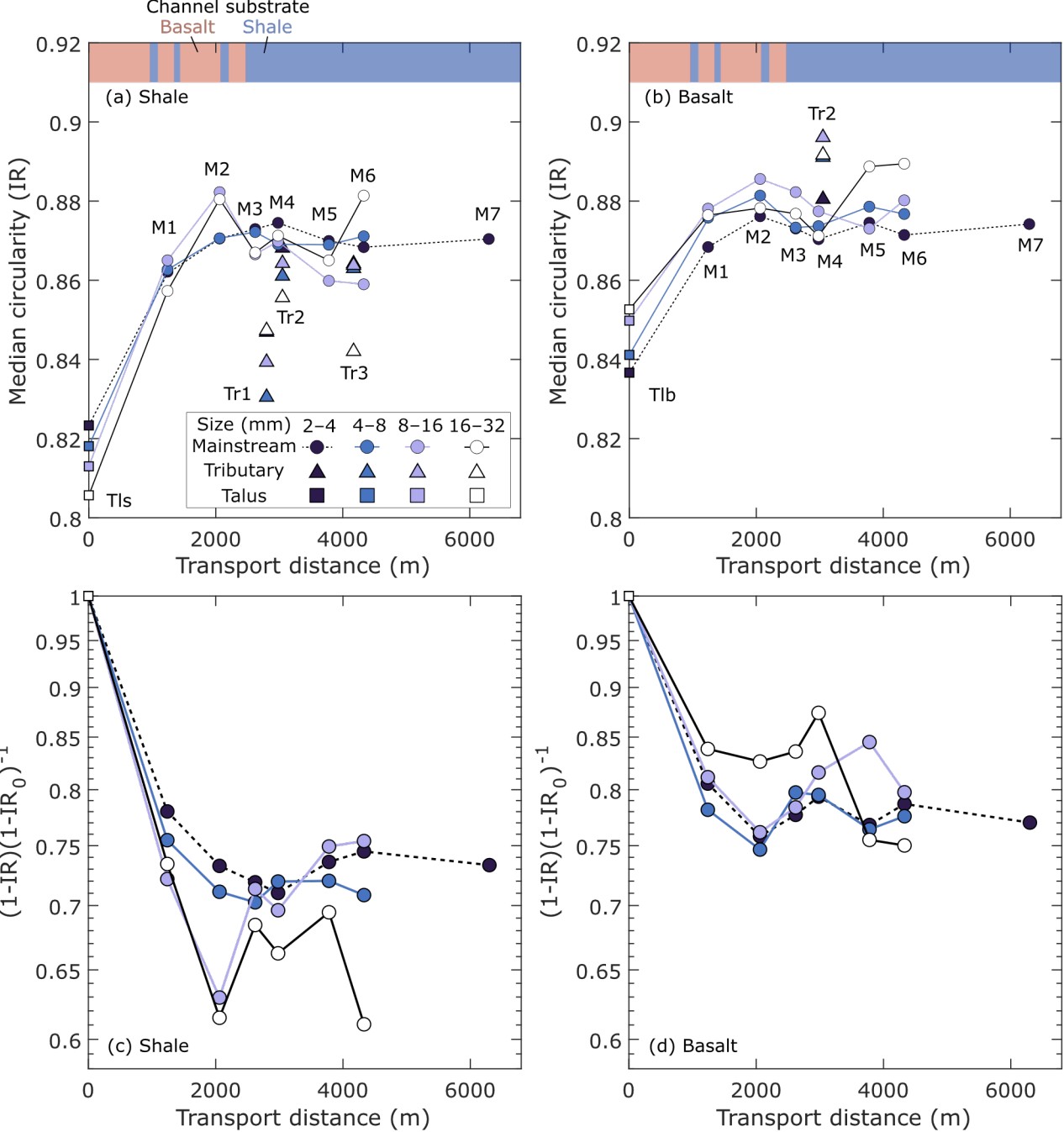

**Figure 5: (a, b) Downstream evolution of circularity (*IR*). (c, d) The ratio of the difference between the limiting *IR* and measured *IR* at each sampling site (*IR*) and channel head (*IR*$_0$). Tributary data are plotted at the confluences with the main stream. Talus data are plotted at the presumed channel heads, not at the actual sampling locations. The vertical axis is in a logarithmic scale.**

**Table 3: Rates of change in isoperimetric ratio ($IR$)**

| Extent | All | | Channel head to M2 (< 2060 m) | | M3 to M7 (2620–6300 m) | |
|---|---|---|---|---|---|---|
| Grain size (mm) | $\lambda_{IR}$ (km-1) | p | $\lambda_{IR}$ (km-1) | p | $\lambda_{IR}$ (km-1) | p |
| Shale | | | | | | |
| 2–4 | 0.04 | 0.085 | 0.16 | 0.137 | -0.01 | 0.306 |
| 4–8 | 0.07 | 0.037 | 0.17 | 0.154 | 0.00 | 0.804 |
| 8–16 | 0.05 | 0.259 | 0.23 | 0.073 | -0.04 | 0.127 |
| 16–32 | 0.09 | 0.034 | 0.24 | 0.024 | 0.05 | 0.376 |
| | | | | | | |
| Basalt | | | | | | |
| 2–4 | 0.04 | 0.081 | 0.14 | 0.125 | 0.00 | 0.464 |
| 4–8 | 0.05 | 0.080 | 0.15 | 0.166 | 0.02 | 0.186 |
| 8–16 | 0.04 | 0.168 | 0.14 | 0.116 | -0.01 | 0.678 |
| 16–32 | 0.06 | 0.010 | 0.10 | 0.215 | 0.09 | 0.126 |

The differences in $IR$ distribution between adjacent sites are presented in Fig. 6 and in Takahashi et al. (2025). Similar to the differential histograms for $R$, the differential histograms for $IR$ in the most upstream section were smooth and had single peaks in the positive and negative change domains (Fig. 6a). The histogram shapes in the other sections were much more complex than those in most upstream sections. When the net change in $IR$ was small, the histogram was jagged and contained multiple peaks in the positive and negative domains (Fig 6b). When the median $IR$ decreased (Fig. 6c), the differential histogram was smoother than when there was little change in the median value $IR$ (Fig. 6b).

Particles supplied from tributaries have different $IR$ values than those in the main stream. For shale, tributaries fed particles of lower $IR$ to the main stream, some of which coincided with a decrease in $IR$ in the main stream, such as between 2980 and 3780 m (Fig. 5a, Tr2 between M4 and M5). In contrast, although 16–32 mm particles introduced to the main stream at 4270 m had a lower $IR$ than those in the main stream (Tr3), the median $IR$ in the main stream significantly increased downstream from the tributary junction (Fig. 5a). For basalt, $IR$ values for 16–32 mm particles increased between 3780 and 4330 m (M5 to M6), where particles of higher $IR$ were supplied from the tributary (Tr2). However, in the same section, $IR$ values for 8–16 mm particles decreased, despite the supply of more circular particles from the tributary.

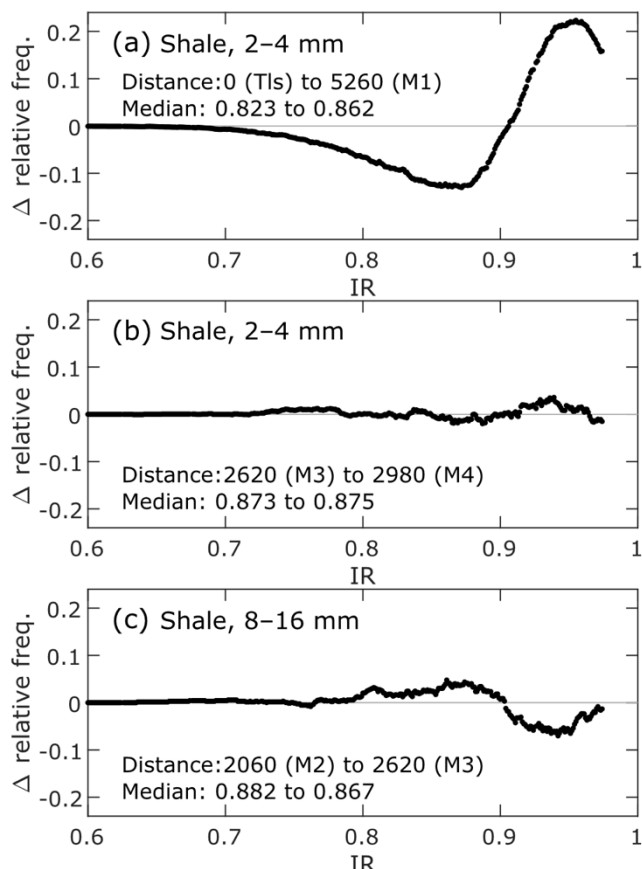

**Figure 6: Changes in *IR* distribution between adjacent sites. The distance between the two sampling sites and their median *IR* values are provided in the lower-left of each panel.**

### 4.1.3 Effect of sample size on the sample median *R* and *IR*

Figure 7 shows the maximum and the minimum difference between the median *R* and *IR* values of particles randomly selected from each of the three samples (M7, Tlb, and M6) and their true median values. The deviation of sample median strongly depends on the sample size and the variance of the population. When the sample size is 100, the sample median *R* can deviate from the true median by +0.013–0.032/-0.037–-0.014 (Fig. 7a). These uncertainty in the sample median decrease to +0.003–0.007/-0.008–-0.003 when the sample size increases to 2000, which roughly equals to the average sample size in this study. For *IR*, the deviation of the sample median is +0.013–0.016/-0.018–-0.014 when the sample size is 100 (Fig. 7b). These values decrease to +/-0.003–0.004 when the sample size increases to 2000.

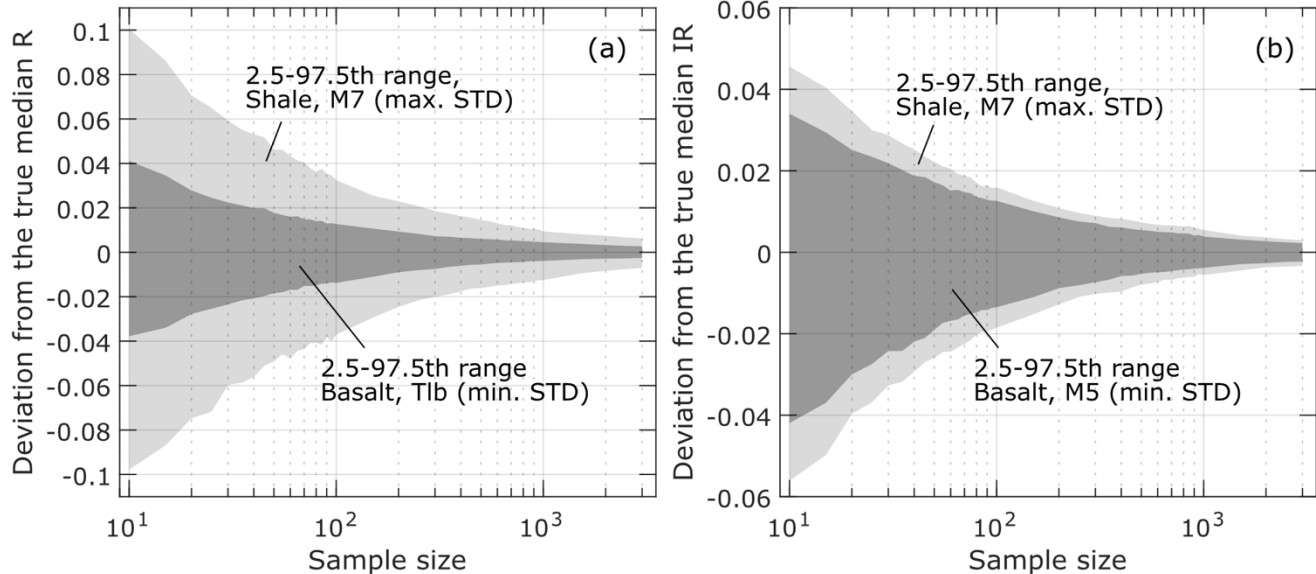

**Figure 7: Deviation of the sample median (a) *R* and (b) *IR* values from the true median values as a function of sample size. The thinner gray area represents the range of 2.5th–97.5th percentile deviation of a sample with the maximum standard deviation of *R* or *IR*. The thicker gray area represents the range of 2.5th–97.5th percentile deviation of a sample with the minimum standard deviation of *R* or *IR*.**

## 4.2 Changes in shape due to manual crushing

The crushed 2–4 mm particles were characterized by lower *R* and *IR* values compared with their original 8–16 mm particles (Fig. 8). The median *R* and *IR* values for the talus samples decreased from 0.403 to 0.390 and 0.813 to 0.797, respectively (Figs. 8a and 8b). The median *R* and *IR* values for the fluvial sample (M6, 4330 m) decreased from 0.550 to 0.393 and 0.859 to 0.795, respectively (Figs. 8c and 8d). Despite the significant differences in the original *R* and *IR* values between the talus and fluvial samples, the distributions of *R* and *IR* values of the crushed particles were almost identical (Figs. 8e and 8f). We also compared the shapes of non-crushed talus samples sized 2–4 mm with the two crushed samples. The distributions of *R* between the non-crushed and crushed particles were similar (Fig. 8e), whereas the distributions of *IR* clearly differed (Fig. 8f).

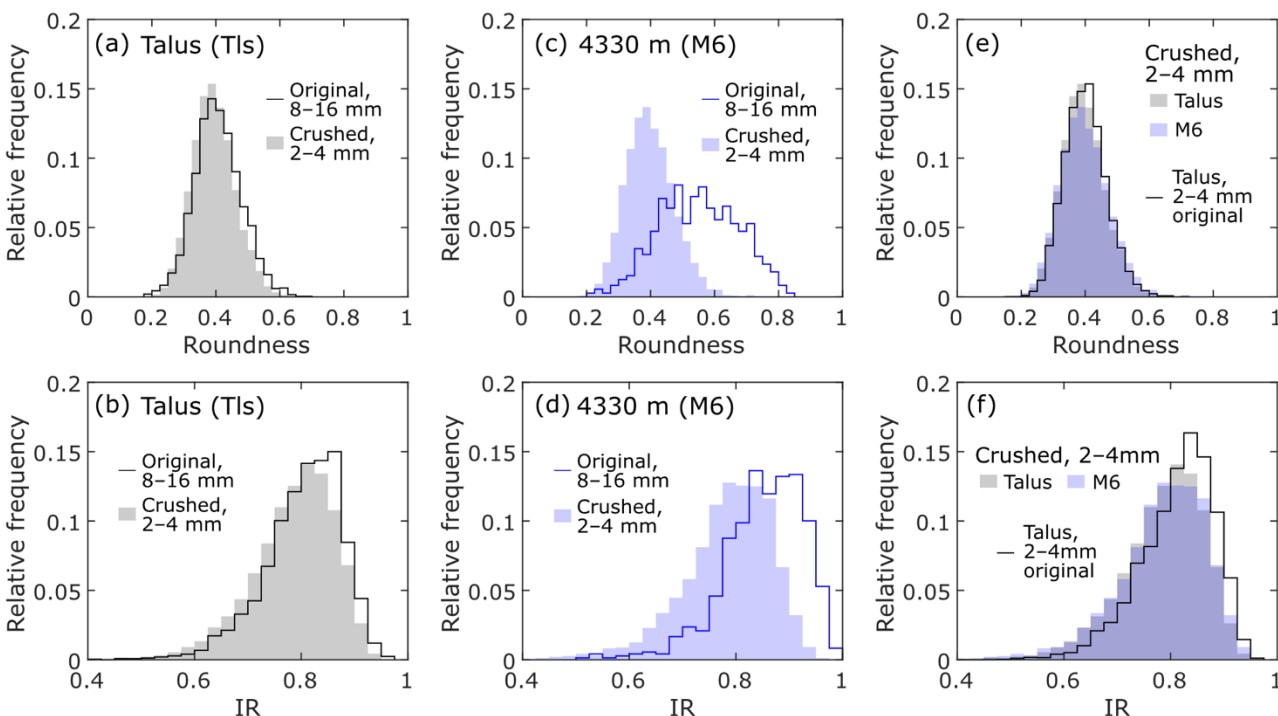

**Figure 8: Changes in *R* and *IR* due to manual crushing. Experimental results using (a, b) talus deposits and (c, d) M6, the 6th fluvial sampling point from upstream. (e, f) Distributions of *R* and *IR* values forthe crushed talus and fluvial samples and non-crushed 2–4 mm talus samples.**


Figure 9 shows the 16th, 50th, and 84th percentile p-values of a Wilcoxon rank sum test performed to examine the difference in shape parameters between the crushed samples and non-crushed talus sample. The p-values for *R* significantly exceeded the common threshold value of 0.05 when the sample size was several hundred and decreased with the sample size (Figs. 9a–9c). This result suggests that the median *R* values of the three samples can be evaluated as different despite the very small

difference. For *IR*, the p-values for the crushed talus–crushed fluvial samples were always much higher than 0.05 (Fig. 9d), suggesting that the median *IR* values of the crushed samples were similar, regardless of the initial distribution before crushing. For the crushed talus–original talus and crushed fluvial–original talus samples, the p-values never exceeded 0.05 (Figs. 9e and 9f), suggesting that the median *IR* values of the crushed samples differed from that of the talus sample.

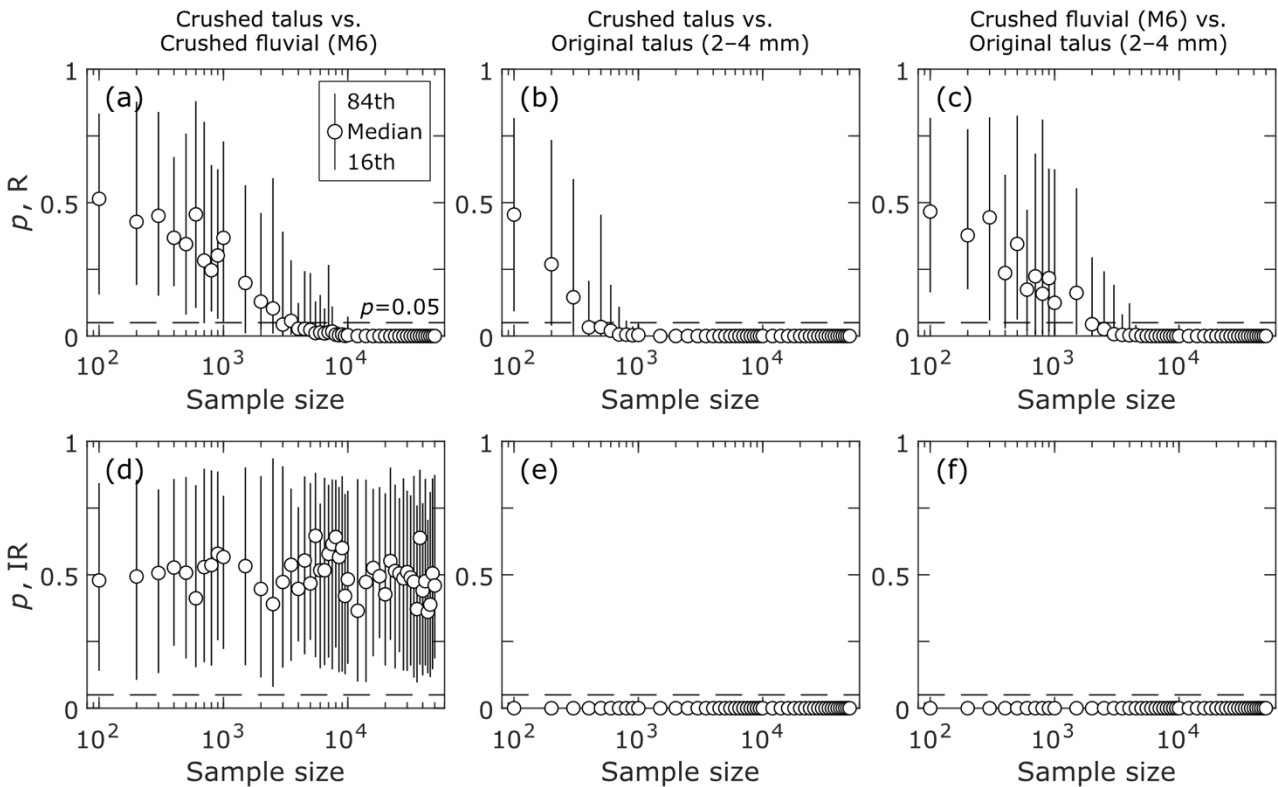

**Figure 9: Impacts of sample size on p-values obtained in a Wilcoxon rank sum test. The samples used in the test are (a, d) crushed talus and crushed fluvial samples, (b, e) crushed talus and original 2–4 mm talus samples, and (c, f) crushed fluvial and original 2–4 mm fluvial samples. The dashed line at the bottom of each panel indicates p = 0.05.**

## 5 Discussion

### 5.1 Downstream shape evolution

The observed changes in *R* and *IR* are clearly different between the upstream and downstream section of the study area (Figs. 3 and 5). The rapid increase in the initial part of transport and the decrease in $\lambda_R$ and $\lambda_{IR}$ with an increase in total transport distance was consistent with the results of other field sites (Wentworth, 1923; Litwin Miller et al., 2014; Novák-Szabó et al., 2018; Ishimura and Hiramine, 2025), theoretical models (Domokos et al., 2014; Novák-Szabó et al., 2018; Pál et al., 2021), and laboratory experiments (Krumbein, 1941a; Kuenen, 1956; Bodek and Jerolmack, 2021; Litwin Miller and Jerolmack, 2021; Bray et al., 2024). In the downstream section, only the median *R* values of basalt samples increased clearly (Fig. 3 and

Table 2). A marginal increase in the shape parameters following rapid increases is less common and has not been reported by studies focusing on the shape evolution of single particle (Domokos et al., 2014; Bodek and Jerolmack, 2021; Litwin Miller and Jerolmack, 2021). While roundness can stop increasing after tens or hundreds of kilometers of transport (Sneed and Folk,

1958; Roussillon et al., 2009), the marginal increase in $R$ in Tsugaru occurred within a few kilometers from the channel head. Thus, identifying the cause of this punctuated increase can potentially provide new insights into the relative importance of relevant processes such as chipping, lateral sediment supply, and weathering.

Given the short length of the studied reach and the limited number of samples, it is important to consider uncertainty in the measured shape parameters. Possible sources of uncertainty include the difference between a sample and its population, local natural variability in particle shape (Rice & Church, 1998), the heterogeneity of hillslope material, and the dependance of shape parameters on projection planes (Buchwald et al., 2025; Tripathi et al., 2025).

We first discuss the difference between a sample and its population. The average number of particles analyzed increased with particle size: 3958 for 2–4 mm, 2605 for 4–8 mm, 1141 for 8–16 mm, and 350 for 16–32 mm (Table 1). When the sample size is greater than 1000, median $R$ and $IR$ values of the sample can differ from the true median values by 0.009–-0.013 and 0.005–-0.006, respectively (Fig. 7). These deviations are smaller than changes in $R$ and $IR$ values between adjacent sites (Figs. 3 & 5), suggesting their impact on the interpretation of the shape evolution are minimal. When the sample size is 350, the deviation of the median $R$ and $IR$ values of a sample is 0.017–-0.019 and 0.008–-0.01, respectively (Fig. 7). In the downstream section (M3 to M7), these deviations are greater than changes in median $R$ and $IR$ values of 16–32 mm particles between adjacent sites.

Local natural variability of particle shape refers to the heterogeneity of particle shape within and between individual gravel bars (Rice and Church, 1998). For the within-bar variability, we argue the individual $R$ and $IR$ values are similarly biased because all the fluvial samples were collected from the edges of a bar near the thalweg and from a similar height relative to the thalweg. For the between-bar variability, we consider the uncertainty using the data presented in Litwin Miller et al. (2014) because we cannot discuss it using our results. They studied the evolution of $IR$ over a 10-km long reach from headwaters based on image-based analysis. In the first 2 km from headwaters, they collected less than 200 particles at 16 sites and calculated their average. The average distance between sampling sites is 125 m, which is much shorter than the sampling interval of our study. The difference in mean $IR$ between adjacent sites is mostly within 0.02, which is likely the sum of the deviation of the sample mean from the true mean, local variability, and the effect of projection.

To estimate the deviation of the sample, we performed the same analysis as we did to estimate the difference between the sample median and true median (Fig. 7). When the sample size is 200, the deviation of sample mean $IR$ was +0.007–0.01/-0.013–-0.009. Based on this result, the total uncertainty excluding the deviation of sample mean from true mean would be about 0.01. As we reported earlier, the sample sizes except for 16–32 mm particles in our study are more than 1000, and the deviation of sample median from true median is ~0.005. Thus, it would be reasonable to assume the total uncertainty in sample median $IR$ is less than 0.02. This maximum uncertainty includes the heterogeneity of hillslope material and the effect of projection. However, the limited influence of tributary supply on the changes in shape parameters suggests local changes in shape of hillslope material does not affect the shape changes of the main stream. This is because the amount of sediment supplied from hillslopes to a particular site in the main stream should be much smaller the amount supplied from a tributary.

Moreover, we argue the effect of projection affects the median $R$ and $IR$ values uniformly and does not alter the overall trend because we aligned the particles on the same basis.

Absolute changes in median $IR$ in the downstream section were smaller than 0.02, suggesting it would be difficult to discuss whether $IR$ did not change or gradually increase. For $R$, we do not have any data or references to assess the local natural variability. However, the observed change tended to follow the exponential rounding model well except for shale samples in the downstream section and 16–32 mm basalt samples in the upstream section (Figs. 3c and 3d). These results suggest uncertainties related to local natural variability and the projection are small compared to the change in $R$ between adjacent sites, or those uncertainties offset the median value of each sample by a similar degree. To further examine if the median $R$ increased in the downstream section, we compared the rate of change in $R$ with those reported by Mills (1979) that compiled data of downstream changes in $R$. Mills (1979) assumed downstream changes in $R$ was a function of a logarithm of transport distance and estimated the rate of changes in $R$ ($\lambda'_R$) using the published data:

$$R = a + \lambda'_R \log x, \tag{7}$$

where $a$ is a constant. Fitting Equation (7) to our result of basalt samples in the downstream section yields $\lambda'_R$ values between 0.05–0.22, which is consistent with $\lambda'_R$ of 0.04–0.26 reported in Mills (1979). Therefore, although the uncertainty in the measured $R$ values cannot be properly estimated, we argue the uncertainty is not large enough to alter the trend of the median $R$ values, and the median $R$ of basalt particles increased in the downstream section.

In the most upstream section (< 2060 m, channel head to M2), $R$ and $IR$ increased much faster than in the rest of the studied section. The transition from initial rapid changes in shape to slower changes occurred 2 km downstream from the channel head, which is consistent with previous findings that a similar transition occurred in the first several kilometers of transport (Wentworth, 1923; Litwin Miller et al., 2014). This rapid change in shape is associated with the removal of sharp corners and edges, which causes a decrease in the curvature of the particle corners (Domokos et al., 2014; Litwin Miller and Jerolmack, 2021). The differential histograms in the most upstream sections are much simpler than those in the further downstream sections (Figs 4 and 6). This result implies that the effect of the rapid removal of sharp corners and edges is much greater than that of other processes, such as the supply from adjacent hillslopes and the production of finer particles with low $R$ and $IR$ values due to chipping and fragmentation.

Following the rapid changes, both shape parameters except for $R$ of basalt particles did not increase significantly, as shown by much smaller values of $\lambda_R$ and $\lambda_{IR}$. The shapes of the differential histograms are much more complex than those in the most upstream sections. This suggests that multiple processes that increase or decrease $R$ and $IR$ occur in this section. Although the definitions of shape parameters differ among studies, a decrease in roundness and circularity with transport distance has been reported in fluvial and beach environments (MacCarthy, 1933; Russel and Taylor, 1937; Pettijohn and Lundahl, 1943; Sneed and Folk, 1958; Frostick and Reid, 1980). These studies ascribed the observed decrease to sorting by particle shape or preferential particle breakdown by fragmentation over frictional abrasion. The supply of angular or less circular particles from tributaries and hillslopes can potentially cause local decreases in $R$ and $IR$. Therefore, identifying the

cause of the punctuated increases in $R$ and $IR$ requires a careful examination of the processes that potentially affect the observed shape evolution.

We first considered the effects of the sediment supply from tributaries and hillslopes. The $R$ and $IR$ values in the tributaries were generally different from those in the main stream near the tributary junctions (Figs. 3 and 5). However, changes in the median $R$ and $IR$ values in the main stream did not correspond with those expected from the differences in $R$ and $IR$ between

the main stream and tributaries. Moreover, the significant decrease in $\lambda_R$ and $\lambda_{IR}$ started upstream of the tributary junctions (Figs. 3 and 5). Thus, we interpreted that the punctuated increases in $R$ and $IR$ were not caused by tributary inputs. For hillslope supply, the sudden decrease in $\lambda_R$ and $\lambda_{IR}$ for shale corresponded with the transition of bedrock from basalt to shale at 2500 m (Figs. 3 and 5). Downstream from this bedrock transition, continuous hillslope supply probably contributed to the marginal increase in $R$ and $IR$ for shale. The decrease in $\lambda_R$ and $\lambda_{IR}$ for basalt occurred at the same location as that for shale.

Because basaltic particles are not supplied from the hillslope downstream from the bedrock transition at approximately 2500 m, other processes that counteract an increase in shape parameters are required to explain the observed evolution of the particle shape.

The introduction of particles from coarser to finer size classes owing to mass loss during transport may affect the distribution of $R$ and $IR$ for particles of finer size classes. Although it is unclear whether significant loss of particle mass occurs during a

few kilometers of transport, the median grain size estimated from the Wolman pebble count in the studied reach decreased from 15 to 8.4 cm between 940 and 4810 m (Takahashi, 2025). In addition, both $R$ and $IR$ values clearly increased in the studied reach. Thus, the grains that lost their mass and fell into a smaller size class could have affected the observed changes in the particle shape. For shale, $R$ and $IR$ were similar among the four size classes and did not show systematic changes in grain size (Figs. 3 and 5). For basalt, the coarser grains were more rounded than the smaller grains (Fig. 3). The $IR$ values for

basalt were similar among the four size classes. These results suggest that the addition of particles from the larger to smaller size classes did not contribute to the punctuated increase in $R$ and $IR$.

The production of finer particles from chipping and fragmentation may reduce $\lambda_R$ and $\lambda_{IR}$ of bulk sediment. The manual crushing experiment showed that the crushed particles had lower $R$ and $IR$ values than the parent particles (Fig. 8). The experiment also showed that the absolute $R$ and $IR$ values of the crushed particles were similar regardless of $R$ and $IR$ values

before crushing. The number of rock fragments produced during the particle mass loss can be much larger than that of the parent particles (Litwin Miller and Jerolmack, 2021; Bray et al., 2024). The particle sizes sampled were much smaller than the median grain size of the riverbed materials (Takahashi, 2025), suggesting that the production of finer fragments affected the median $R$ and $IR$ values.

For shale, coarse particles with lower $R$ and $IR$ values were continuously supplied from adjacent hillslopes in the middle

section. These particles likely produced many fragments during the initial stage of mass loss, lowering $\lambda_R$ and $\lambda_{IR}$ in the downstream part of the studied reach. Basaltic particles were not supplied to the main stream downstream from the bedrock transition at approximately 2500 m, suggesting that their $R$ and $IR$ values were less affected by the production of rock

fragments than those for shale particles. This interpretation is supported by the fact that the downstream increase in median $R$ values for basalt started at 2980 m (M4), whereas that for shale started further downstream (Fig. 3). Nevertheless, basaltic

gravels constituted at least 64% of the bed materials in the reaches lacking supply of basalt particles from valley-side slopes (Takahashi, 2025), suggesting an abundant presence of basaltic grains that can produce fragments. Moreover, while $R$ values of crushed samples are similar to those of talus samples, $IR$ values are different between the crushed and talus samples (Fig. 8). Thus, we argue that the production of finer fragments also contributed to the relatively constant $IR$ of the basalt particles in the downstream section of the study area.

The upstream end of the punctuated increase in $R$ and $IR$ corresponds to a significant decrease in the channel slope, suggesting a change in transport capacity. As in the case of the gravel–sand transition (Dingle et al., 2021), the decrease in transport capacity can change the impact of particle sorting on the composition of bed materials, increasing the proportion of finer and more mobile particles. Previous studies (MacCarthy, 1933; Pettijohn and Lundahl, 1943; Frostick and Reid, 1980) attributed the decrease in roundness or sphericity with transport distance to the effects of shape-sorting because saltating or

suspended particles of more irregular shapes can outpace more spherical particles owing to greater drag force (Deal et al., 2023) and smaller settling velocity (Briggs et al., 1962; Dietrich, 1982; Ferguson and Church, 2004). In contrast, when particles roll or slide on a bed, spherical particles tend to travel longer distances than non-spherical particles (Hattingh and Illenberger, 1995; Cassel et al., 2021). Thus, to evaluate whether shape sorting played a role in this study, the dominant transport modes of the sampled particles must be identified.

The finest size range analyzed in the study was 2–4 mm, close to the suspension threshold during typical flood flows in alluvial rivers (Parker et al., 2024). The sizes of the sampled particles were significantly smaller than the median grain size of the bed material (Takahashi, 2025), suggesting that these finer particles were often sheltered by the surrounding grains. The critical Shields number increases nonlinearly with decreasing grain size relative to the size of surrounding grains (Hodge et al., 2020; Smith et al., 2023). Thus, when the measured fine particles are entrained, their transport stage, which is the ratio

of the Shields number to the critical Shields number without the sheltering effect, is likely well above unity.

Considering that the probability of transport by rolling decreases exponentially with the excess transport stage (Auel et al., 2017; Demiral et al., 2022), most of the particles studied were probably saltated or suspended during transport. Therefore, we interpret that the effects of shape sorting partly explained the observed decreases in $R$ and $IR$. However, this interpretation is admittedly expedient because some studies have shown that the particle shape does not significantly affect

the transport distance of saltating particles (Chatanantavet et al., 2013; Demiral et al., 2022). Nevertheless, we argue that the decrease in $R$ and $IR$ in the present study is difficult to explain without the effect of shape sorting because both shale and basalt particles showed a similar decrease at the same location despite their differences in spatial distribution and material properties.

In the most-downstream section, where the median $R$ of basalt slowly increased, the differential histogram was similar to

that of the most-upstream section (Fig. 4). Although the absolute difference in $R$ was much smaller than that in the most-upstream section, this result indicates that the combined effects of processes that increase roundness likely prevail over those

that decrease roundness. One possible reason for this resurgence of rounding is that the impacts of lateral supply and chipping- and fragmentation-derived rock fragments decrease downstream as the total amount of bed material in the main stream increases. Assuming the number of chipping- and fragmentation-derived fragments decreases as the particle wear

proceeds (Bray et al., 2024), the total number of newly added particles per transport distance initially increases downstream and reaches a constant value at some point. In this case, the ratio of new particles to existing grains initially increases and then keeps decreasing. Therefore, without significant sediment supply from tributaries or mass movement, we infer the general rounding process dominate shape evolution of the bed material except for a short river section near a channel head.

## 5.2 Factors controlling particle shape

The most evident factor affecting $\lambda_R$ and $\lambda_{IR}$ was material properties such as hardness and brittleness (Attal and Lavé, 2009; Bodek and Jerolmack, 2021). Values of $\lambda_R$ and $\lambda_{IR}$ were larger for shale than for basalt (Tables 2 and 3). The faster rates of mass loss of the shale particles compared with basalt particles likely resulted from their lower mechanical strength and higher susceptibility to slaking. Most of the shale particles found in the riverbed were strongly weathered and could be easily

crushed manually. In addition, cracks can easily form in the shale particles owing to slaking, making the shale particles susceptible to fragmentation.

The values and changes in $R$ of the basalt particles differed significantly by particle size (Fig. 3). The increase in $R$ with particle size was possibly due to the longer residence time of the coarser particles in the channels (Pettijohn, 1949). In addition, as the particle mass decreases, the effects of viscous damping become increasingly important, hampering the mass

loss due to particle collisions (Schmeeckle et al., 2001; Jerolmack and Brzinski 2010). Moreover, the number of finer fragments constitutes greater proportions of the chipping and fragmentation products (Domokos et al., 2015), thereby reducing the median $R$ of finer particles. While we interpret these three factors are responsible for the observed increase in $R$ with particle size, it is also true that shape parameters do not always increase with particle size (Russell and Taylor, 1937; Sneed and Folk, 1958; McBride and Picard, 1987). In fact, median $R$ values of shale particles did not change with the size

classes, which we will discuss later. Also, our interpretation may be incompatible with the smallest $\lambda_R$ for the largest basalt particle (Table 2). We argue that this conflict is related to weathering processes on hillslopes.

The weathering of igneous rocks typically occurs along cooling joints, which eventually produces well-rounded corestones due to spherical weathering (Hirata et al., 2017). However, finer particles produced by weathering detach from coarser particles and experience weathering for a shorter duration, suggesting that they have relatively fresh edges compared with

coarser particles. Although rock failures or deep-seated landslides can supply unweathered gravel to channels, large-scale colluvial processes are probably uncommon in the study area because the average angle of hillslopes composed of basalt (distance > 4000 m) was 24.8°, which is smaller than the typical threshold angle above which landslides dominate hillslope denudation (Roering et al., 2007; Larsen and Montgomery, 2012; DiBiase et al., 2023). In addition, our field observations confirmed that most slope failures were shallow and did not supply many boulders to the channels. Therefore, we interpreted

that the roundness of the talus deposits of basalt increased with the particle size because of the greater influence of

weathering. This resulted in the removal of the angular corners of coarser particles that were otherwise removed at the initial stage of the fluvial transport, leading to a marginal increase in roundness in the headwaters.

While the median $R$ value of basalt clearly depended on particle size, the median $IR$ of basalt did not (Figs. 3 and 5). This is because the maximum $IR$ that a particle can achieve increases with aspect ratio, which is the ratio of the intermediate ($b$) to the longest ($a$) axis length (Takashimizu and Iiyoshi, 2016; Quick et al., 2020). The median aspect ratio of the basalt particles decreased with particle size: 0.749, 0.744, 0.739, and 0.731 from the 16–32 to 2–4 mm sized particles. To determine the impact of the aspect ratio on $IR$, we corrected the effect of the aspect ratio on the measured $IR$ values by diving the $IR$ of the particles by the $IR$ of a perfect ellipse with the same axis ratio $IR_t$ (Villarino, 2005; Quick et al., 2020; Pokhrel et al., 2024). The calculated $IR/IR_t$ values increased with the grain size (Fig. 10), indicating that the insensitivity of $IR$ to the size of the basalt particles was mainly due to the different aspect ratios among the particle sizes. In addition, the downstream change in $IR/IR_t$ was consistent with that of $R$, confirming the rationale of Quick et al. (2020) that $IR/IR_t$ represents particle roundness.

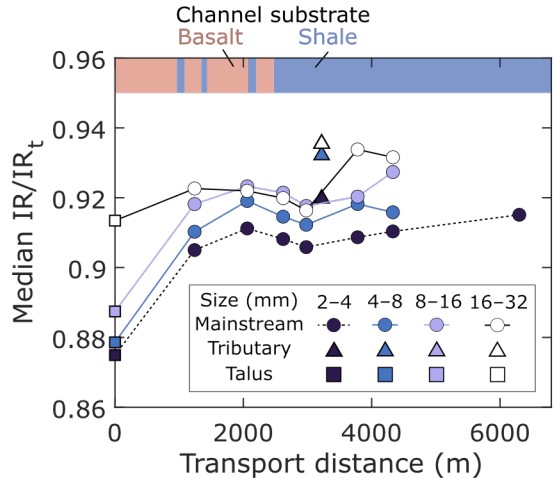

**Figure 10: Downstream evolution of $IR/IR_t$ for basalt samples. Tributary data are plotted at the confluences with the main stream. Talus data are plotted at the presumed channel heads, not at the actual sampling locations. The vertical axis is in a logarithmic scale.**

The roundness of the shale particles did not clearly change with the grain size (Fig. 3), which is attributed to the rapid weathering of the shale. Sak et al. (2010) investigated the thickness of the weathering rinds of volcanic rocks and found that the rates of weathering rind formation were higher in areas of higher curvature at the rind–core interface. As in the definition of Wadell's roundness (Eq. 1), corner curvature of particles, an inverse of a radius of curvature of a corner, with the same $R$ increase with decreasing particle size, indicating that rind formation may be faster for finer particles. Even if the finer and coarser particles have the same rind thickness, as assumed by Jones and Humphrey (1997), the areal or volumetric

proportion of the rind within a particle was larger for finer particles, suggesting that weathering has a greater impact on mass loss. Although we did not conduct a chemical analysis of the basalt particles, the lack of a visible weathering rind of basalt particles in the riverbed suggests that such an effect of weathering does not apply to the basalt particles measured in this study.

**5.3 Limitations of this study**

     Although the current analysis enabled the examination of the downstream evolution of particle shape and its rates, it also presents challenges that require further study. We focused on the median $R$ and $IR$ values for simplicity; however, more information could potentially be obtained from a differential histogram. For instance, basalt particles with $R > 0.55$ clearly increased from 4330 to 6300 m (Fig. 4d, M6 to M7). The increase in the relative frequency of $R$ was roughly constant

between $R$ values of 0.55 and 0.74, which may be incompatible with Eq. (3) that predicts the rate of increase in $R$ decreases with increasing $R$ (Krumbein, 1941a; Novák-Szabó et al., 2018). We speculate that this is related to the supply of well-rounded particles from alluvial storage (Jones and Humphrey, 1997). The shapes of individual histograms, such as the modality, skewness, and kurtosis, may also provide additional insights into the control of flow and material properties on shape evolution. However, studies under more controlled conditions are required to fully interpret the distribution of shape

parameters, because we do not know how various processes operating in natural rivers affect the distribution of shape parameters.

     We did not consider the effects of particle loss during transport (Sklar et al., 2006; Mueller et al., 2016). If the sediment flux depends on particle shape, fine particles of a certain shape may be preferentially transported over longer distances than other particles and exported out of the catchment, potentially affecting the distribution of shape parameters for finer particles.

The particles we investigated were much smaller than the median size of the bed materials, suggesting that our findings may not be applicable to studies on particle size and mass loss in natural rivers that usually focus on median-sized or even coarser grains (Rice and Church, 1998; Attal and Lavé, 2006). The size dependence of the changing rate of shape parameters suggests that inferences on particle mass loss based on particle shape can also differ according to particle size. Although image-based analysis of particle shape has enabled the efficient measurement of riverbed gravel, analyzing the shape of

coarse gravel typically found in gravel-bed rivers in steep mountains is still difficult. Given the growing recognition that a coarser fraction of bed material strongly affects the erosion rates and morphology of mountain rivers (MacKenzie et al., 2018; Shobe er al., 2021; Sklar, 2024; Takahashi, 2025), further studies on the size dependency of particle shape evolution are required to link the mass loss of fine particles to the evolution of fluvial landscapes.

     We used transport distance, the horizontal distance from a channel head, as an independent variable in Eqs. (3) and (4).

While transport distance is commonly used to study downstream changes in the size and shape of sediment (Sternberg, 1875; Krumbein, 1941a), it is challenging to measure the actual transport distance of single particle. This is because a channel centerline estimated from DEM is much simpler than the actual centerline, and the bed sediment is a mixture of particles with various transport histories. In addition, given the importance of particle mobility in the evolution of fluvial landscape

(Mackin, 1948; Sklar and Dietrich, 2006; Parker et al., 2007), mass loss of a particle due to changes in shape should also be studied. Using tracer particles can help measure changes in particle mass in the field (Novák-Szabó et al., 2018). However, it would be difficult to retrieve hundreds of tracer particles after several kilometers of transport that is required to cause measurable changes in particle shape.

Lastly, the difference between the 2D and 3D shape parameters is important to leverage our findings to understand the shape evolution of single particle and evaluate impacts of sediment load on channel morphology. Some shape parameters have a 3D counterpart (e.g., *IR* and sphericity). 2D parameters have been used more widely due to relative ease of measurement (Tripathi et al., 2025). However, 2D parameters are sensitive to particle orientation, which limits the use of 2D parameters as a proxy for 3D particle shape (Buchwald et al., 2025; Tripathi et al., 2025). Bagheri et al. (2015) measured 3D shape of volcanic ejecta and examined an effect of particle orientation by calculating *IR* for a thousand projection planes generated from each particle. They showed that while *IR* of the maximum projection plane can deviate from the mean *IR* of the 1000 planes by about 10% of the mean *IR*, suggesting care must be taken when interpreting absolute values of 2D shape parameters. We argue that increasing sample size can reduce uncertainties in measured shape parameters associated with particle orientation and natural variability, which enables to obtain a representative value of 2D shape parameters. Nevertheless, it is important to study the correlation between such a representative value of 2D shape parameters and 3D particle shape.

## 6 Conclusion

This study measured the particle shapes of talus deposits and riverbed materials using an automated image analysis tool and examined the downstream changes in particle roundness and circularity and their rates of increase with transport distance. The upstream part of the study area were characterized by a clear increase in roundness and circularity. In the downstream section, the median roundness of basalt particles increased downstream at a slower rate than in the upstream section. The median roundness of shale particles and median circularity of shale and basalt particles remained almost constant in the downstream section owing to the effects of the hillslope supply and the chipping- and fragmentation-derived finer particles counteracting the increase in both shape parameters. Although chipping and fragmentation should affect the median roundness of basalt particles in the downstream section, their impacts were not significant enough to retard the increase in the median roundness. Regarding the effects of rock type and particle size on the changing rates of shape parameters, the rates for shale particles were faster than those for basaltic particles and did not depend on the particle size. In contrast, the rates for coarser basalt particles were higher than those for finer particles. These differences are attributed to the weathering characteristics in the channel and on the hillslopes, the mechanical strength of the particles, and the residence time in the channel. Our findings indicate that because of the significant changes in shape parameters over a short transport distance, headwater streams are suitable for evaluating the contributions of individual processes and material properties to

downstream changes in shape parameters. Nevertheless, further studies that relate changes in particle shape to understand the impacts of shape evolution of bulk sediment on long-term landscape evolution.

## Data availability

All particle shape data are available in figshare at http://doi.org/10.6084/m9.figshare.28424138

## Author contributions

Conceptualization: NOT. Formal analysis: NOT and DI. Funding acquision: NOT and DI. Investigation: NOT, DI, RO, YA and YY. Software: KY. Writing – original draft preparation: NOT. Writing – review & editing: All authors.

## Competing interests

The authors declare that they have no conflicts of interest.

## Acknowledgements

We thank Mikael Attal and Prakash Pokhrel for their helpful discussions. The color maps used in Figs. 3, 5, and 10 were obtained from Crameri (2023). The author would like to thank Editage (www.editage.jp) for English language editing.

## Financial support

This research was financially supported by Japan Geographic Data Center and JSPS KAKENHI (Grant Numbers JP21H00631 JP24K00173, and 22KJ2776).

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
