# Peer review of "Shape evolution of bulk sediment in headwater streams: effects of rock type and particle size"

_EGUsphere, 2025_

## Author Response (AR1)

Response to RC1

[AR] Thank you very much for taking time to evaluate our manuscript and providing constructive comments. We believe the manuscript has been greatly improved based on your suggestions. Changes made based on comments from the other reviewer are also significant, so please consider reading our response to RC2 if necessary. The main points related to comments from the other reviewer include (1) changing the focus of the manuscript from "three-stage evolution" to "shape evolution and its causes" and (2) uncertainty in the measured shape parameters.

General comments

The manuscript "Three-stage evolution of particle shape in headwater streams" investigates the shape evolution of sediment particles along the Mosawa River in NE Japan. Both basalt and shale particles are studied, with a focus on the evolution of their circularity and roundness. The paper's topic and findings definitely fit into Earth Surface Dynamics. Still, the results' presentation and discussion need improvement before the final publication.

The most crucial issue is that both roundness and circularity are computed for 2D images taken of spatial particles. Please discuss the limitations of this approach. Especially the sentence in line 169 "···we aligned the particles with their maximum projected areas parallel to the tray···" should be justified: one would expect, that the particles sit on one of their stable balancing points (a.k.a. equilibrium point), which is not needed to yield their maximum projected area.

[AR] Line 202: We added paragraphs discussing limitations. Regarding the sentence "···we aligned the particles with their maximum projected areas parallel to the tray···", we added a supplementary sentence:

"However, in practice, most natural particles cannot stand still with their maximum projected areas upward. In such cases, we used a particle facet that a particle can stand still and with the maximum projection areas among all facets."

Another questionable approach of the manuscript lies in using the transport distance $x$ as a main variable in equations (2)-(4) because $x$ lacks information about the vertical section of the river. It seems, mass loss would be a more natural measure to describe the stage of shape evolution. Please discuss these issues.

[AR] Line 640-: Although we agree that mass loss would be a more appropriate variable to discuss the shape evolution, we used transport distance because we could not track changes in particle mass of individual particles. Instead, we mentioned this point in the section "5.3 Limitations of this study".

Specific comments

Line 65 on page 3: "··· field studies have shown that the rate of change in particle shape decreases with the transport distance···." - this sentence should be clarified: the rate of change of some shape measure (like the axis ratio, the volume··· etc.) can decrease, not the shape itself. Furthermore, a reference to Sternberg's law stating exponential decay of the particle mass/volume is missing here.

[AR] Line 75: Changed the sentence as suggested and cited Sternberg (1875).

Line 71 on page 3: The overview of image analysis is not complete; a recently invented scanning technique for sedimentary particles has been published here:

Fehér Eszter, Havasi-Tóth Balázs, Ludmány Balázs: Fully spherical 3D datasets on sedimentary particles: Fast measurement and evaluation CENTRAL EUROPEAN GEOLOGY 65:2 pp. 111-121. (2022)

[AR] Line 81: Cited examples of 3D shape analysis (Fehér et al., 2023 (same article as you suggested); Tripathi et al., 2025) and modified the corresponding sentences accordingly.

In the first paragraph in section 2.2 on page 5: the formula defining the roundness should be provided, similarly to the formula of the isoperimetric ratio. How is the 'curvature of the corners' defined exactly?

[AR] Line 139: Added a formula defining the roundness and the definition of corners. Method to identify corners and its circle of curvature is shown in Zheng and Hryciw (2015) and summarized in a GitHub page (Yamada, 2024). We cited Yamada (2024) in the method section.

In line 185 on page 7, the workflow in software Rgrains is cited. What about placing it in an appendix or supplementary material to have a complete overview of your work?

[AR] Line 186: We modified method section to clarify the workflow. We also added a figure explaining the workflow (Figure 2). The core part of the analysis to calculate shape parameters is based on an algorithm developed by Zheng and Hryciw (2015). So, we clarified in which part we used an algorithm of Zheng and Hryciw (2015). A summary is already presented in a GitHub page (Yamada, 2024), so we cited Yamada (2024) after mentioning the related process.

Line 378 on page 18: In the discussion authors state "···the differential histograms in the most upstream sections are···that the effect of the rapid removal of sharp corners and edges···". This observation agrees well with (Domokos et. al., 2014) and (Novák-Szabó et. al., 2018); here, a quantitative comparison to their results (based on the average mass loss) would be insightful. Actually, it seems that this first phase observed along the upstream section coincides with Phase-1 introduced by (Domokos et. al., 2014) for individual particles in curvature-driven abrasion.

[AR] Line 673-: We agree that the initial rapid change in shape we observed is similar to Phase-1 proposed by Domokos et al. (2014). We also agree that it is worth looking at downstream changes in particle mass; however, we have already returned some samples to the river due to logistic reason and can only get an incomplete results even if we measure weight of individual particles. Moreover, since we collected samples using a sieve and are unable to see mass loss of individual particles, it is unclear what kind and magnitude of uncertainties are included in the measured mass loss. Alternatively, we mentioned the importance of measuring particle mass and 3D shape parameters to understand the shape evolution.

Line 385 on page 18: The statement "···a decrease in shape parameters with transport distance has been reported in fluvial, beach, and aeolian environments···" is vague: which parameters are you talking about? Actually, fluvial and aeolian environments are rather different; for instance, the isoparametric ratio typically increases in fluvial, but decreases in aeolian environments.

[AR] Line 505-: We changed "shape parameters" to roundness and circularity. Roundness used in the cited studies include those determined visually (using a visual chert) and numerically (same as R in our study). So, we wrote roundness instead of R. We cited aeolian studies as an example of downstream(down-current) decrease in shape parameters. After reconsideration, we found adding aeolian examples was not very helpful, so we removed Eisma (1965) and Winkelmolen (1971).

Lines 460-465 on page 20: These statements, I think, are not supported by the paper. The evolution of R and IR versus the evolution of the mass/volume is a very delicate question, especially because the measurements are carried out on 2D images. Either show a simple model that supports your claims or rephrase this last paragraph.

[AR] You are right. We disregarded the difference between 2D and 3D shape parameters (IR, in particular). We removed the paragraph.

Lines 545-550 on page 23: It would be nice to have some estimate of where the fragmentation ends along the river. Does it coincide with the end of the intermediate phase?

[AR] Line 693: Although we cannot identify where the chipping and fragmentation ends, we could say their impact is minor in the third phase where R and IR slowly increase. So, we added the following sentences.

"Although chipping and fragmentation should affect the median roundness of basalt particles in the downstream section, their impacts were not significant enough to retard the increase in the median roundness."

Minor Comments:

'Roundness' and 'circularity' are not defined in the abstract. I think a short sentence about their difference would improve the text, e.g., a hint that circularity is measured using the entire perimeter, and roundness focuses on the vicinity perimeter segments with high curvature (i.e., edges).

[AR] Line 21: Added the following sentence:

"Roundness is calculated using curvature of particle edges, while circularity is calculated using the entire perimeter of a particle."

Figure 1: What is the meaning of the black dots on panel (a)? The sampling sites should be denoted by the same signs in panels (a) and (b). The starting point of the distance measurement (i.e., the 0 of the horizontal axis on panel (b)) should also be marked on panel (a).

[AR] Line 131, Fig. 1: The black dots indicate sampling sites for talus deposit. We modified the panels (a) and (b) following your suggestions. We also added the location of presumed channel head (transport distance=0).

Before eq. (3) The correct equation reference reads eq. (2).

[AR] Line 156: Thank you for finding this error. We corrected it.

Writing real fractions (instead of power (-1)) would increase the readability of the paper (e.g., in equations (4) and (5)).

[AR] Line 162: You are right, we corrected. We misunderstood the formatting guide of the journal.

Line 468 on page 20: "particle durability" is not well-defined. The material composition, which determines the hardness and the brittleness of the material, seems to be the right framework here.

[AR] Line 586: We changed to "material properties such as hardness and brittleness". We also relpaleced durability with material properties in other parts of the manuscript.

Figure 9: It seems the vertical axis is not on a logarithmic scale, as opposed to the caption of the figure.

[AR] Line 625, Fig. 10: We checked the figure axis again and found it was on a logarithmic scale. The value range of y-axis was so narrow that it looked like a linear scale. It truned out the vertical axis would be better on a linear scale, so we changed it accordingly.

Among the references: (Szabó et. al., 2018) should read (Novák-Szabó et. al., 2018).

[AR] Corrected to Novák-Szabó throughout the manuscript.

Response to RC2

General comments:
This manuscript presents new data on particle roundness and circularity from a small catchment in the Tsugaru Mountains of north-eastern Japan. The study applies previously established roundness and circularity metrics to analyse the downstream evolution of sediment shape over a 6 km transport distance along the main trunk stream. Using multiple shape metrics, the study evaluates which parameters show significant downstream changes. In addition, the authors compare the shape distribution of fine sediment fractions (2–4 mm) between talus-derived crushed material and downstream fluvial samples. The structure and title of the manuscript suggest that the main finding is a three-stage evolution of particle shape within the headwater stream: an initial rapid increase in roundness, a middle phase of minimal or no change, followed by a gradual increase downstream.

Previous theoretical models supported by field evidence (e.g., Krumblein 1941; Domokos et al., 2014; Miller et al., 2014; Roussillon et al., 2009; Pokhrel et al., 2024) have consistently demonstrated a non-linear, two-phase downstream evolution of roundness that reflects the exponential nature of the abrasion process. These models predict a rapid increase in roundness in the upstream reaches followed by a slower change downstream, with the rate of change dominantly controlled by lithology.

In contrast, the present study presents a three-stage model based on field data from six locations spanning approximately 6 km (roughly one sample per kilometer). However, this is a relatively short segment compared to previous studies that examined roundness evolution over tens to hundreds or even thousands of kilometers. It is likely that the observed three-stage pattern in this study reflects local variability, sampling resolution, or natural scatter in the roundness data—possibly due to heterogeneity in source material—rather than a distinct or generalizable evolutionary model.

Importantly, the manuscript does not provide a functional explanation or mathematical formulation for the proposed three-stage trend, nor does it convincingly argue for its broader applicability. The manuscript lacks a clear explanation of the mathematical expressions or functional combinations used to fit the trend line to the field data, which forms the basis for the author's conclusion of a three-stage evolution in the headwater region. As a result, the primary conclusion—that of a three-stage evolution—is not sufficiently supported by the data or linked to a broader theoretical framework.

Therefore, I do not find adequate scientific justification for the main claim of a distinct three-stage evolution in this study. In its current form, the manuscript does not meet the threshold for publication.

That said, the study does provide useful insights into:
initial distribution of shape parameters from the talus sediment
which shape parameters change most significantly over short distances, and
how particle shape distributions differ when influenced by different mechanisms (e.g., talus vs. fluvial transport).

These findings have potential value and could form the basis of a new manuscript focused more clearly on shape parameter comparison and the mechanics of particle abrasion.

Repply to general comments
[AR] Thank you very much for taking time to evaluate our manuscript. We appreciate your detailed feedback. We have reconsidered the uncertainties of the shape parameters and found that it would be better to simplify the focus of this study rather than proposing the three-stage evolution model. Although the content is essentially the same, we stopped stating the three-stage evolution. Nevertheless, based on the analysis of the uncertainties, we still think the observed trend and its relation to particle size and rock type is meaningful. While most studies measure shape of 100 or less particles, we measured hundreds to thousands of particles per site. Also, we took samples at a much shorter intervals than most of the previous studies. It is true that many of the previous studies focused on shape evolution over much lon

ger distance, but shape parameters can reach a limiting value and do not change over the significant portion of their study area (Russel and Taylor, 1937; Krumbein, 1942; Plumley, 1948; Sneed and Folk, 1957; Roussillon et al., 2009), Moreover, the absolute change in shape parameter in the previous studies is comparable to the absolute change we observed probably because talus data are not included (Krumbein, 1942; Plumley, 1948; Sneed and Folk, 1957; Roussillon et al., 2009; Pokhrel et al., 2024). For these reasons, we shifted the focus of the manuscript to the shape evolution and its cause while keeping the interpretation of the results similar to the initially submitted manuscript.

Regarding the comment on local variability, sampling resolution, or natural scatter in the roundness data, we performed an additional analysis and compared the expected unceratinties and the observed changes in shape parameters to discuss if the observed trend is meaningful.

Regarding the comment on a functional explanation or mathematical formulation of the proposed model, we do not think it is possible to propose a reliable model at this stage. This is because the shape evolution we found applies only to shape changes in bulk sediment, and it requires many unconstrained variables such as the production rate of rock fragment due to chipping and fragmentation, the decaying rate of such production rate as particles lose mass, and the effect of sorting on shape evolution. However, it is true that we failed to clarify what we focused on was the shape change in bulk sediment rather than single particle, which we have addressed in this revision. Also, similar to the irrgular pattern of shape evolution, downstream change in grain size does not always follows the Sternberg's law. Many studies have reported such irregular changes in grain size without developing a functional explanation or mathematical formulation, but those studies are still valuable.

Specific comment
Abstract
Line 14: The phrase "each process" is used without having mentioned any specific processes earlier. Please define or list the processes first before referring to them.
[AR] Line 17: We listed the processes that we focused on.

Line 16: It would be clearer to include the term "downstream" to specify the direction or context of evolution being discussed.
[AR] Line 23: We added "downstream" to the sentence.

Introduction
Several references are missing where key information is provided. For example, the statement in lines 55–56 requires a proper citation.
[AR] Line 64 and other parts: Added references where necessary.

Line 87: Instead of saying "various processes", it would be clearer to name the specific processes and indicate the shape metrics or indices used as proxies to assess them.
[AR] Line 98: We simplified the corresponding part as follows: We aimed to reveal (1) what processes cause a downstream change in roundness (Wadell, 1932) and circularity (Cox, 1927)⋯

Line 127: The manuscript mentions the use of Wadell's roundness and briefly explains its definition but fails to provide the mathematical expression. Readers would expect this expression in Equation 1, rather than introducing the circularity index first. It is recommended to present all shape metrics in sequence, accompanied by their respective mathematical expressions.
Line 141 / Equation 2: The metric 'R' is used (citing Krumbein, 1941a), but there is no explanation of what 'R' represents or how it is measured. If 'R' refers to Wadell's roundness, this should be explicitly stated and described using the proper formula, similar to the explanation given for IR in Equation 1.

[AR] Line 139: We added an equation describing Wadell's roundness (R) as Equation 1.

Figures 1a and 1b: Assign unique labels to each sampling site and use these consistently throughout the results and discussion sections.
[AR] Line 131: Assigned unique labels and modified the corresponding parts throughout the manuscript.

Figure 1c: This subplot shows slope versus contributing area, but the relevance of this plot is unclear. Please explain its purpose and where it is discussed in the main text.
[AR] Line 131, Fig. 1: We included the subplot to show the basic information of the studied river. We agree that the relevance was unclear, so we replaced the subplot with field photographs to explain conditions of the sampling sites.

Methods
Sampling
Line 155: The purpose of collecting samples from the trunk stream and talus is explained, but the rationale for sampling from tributaries and downstream of the dam is lacking and should be elaborated.
[AR] Line 170-: We explained why we took samples from tributaries and downstream of the dam. For the sample downstream of the dam, we took the sample simply because it would be better to see changes in shape over longer distances. We did not take sample further downstream of the current study area because the further downstream section is surrounded by paddy field.

The number of samples collected at each sampling site for each size fraction should be clearly described in this section, instead of mentioning it later in the image analysis section.
[AR] Line 183, Table 1: We added a table of sample size and moved the corresponding part from the image analysis to the method section.

Image Analysis
The manuscript lists the use of RGrains, MATLAB v4.11, Adobe Photoshop, and Affinity Photo—none of which are open-source tools. To improve transparency and reproducibility, the authors are encouraged to include a clear workflow diagram outlining each step from the raw field photograph to the final shape parameter extraction. This would clarify the role and purpose of each software used. For instance, line 168 states, "We used MATLAB application v4.11", but does not specify the function or task it was used for.
[AR] Line 186-, Fig. 2: We added a figure outlining the workflow and explained more about the tools and functions used in this study. MATLAB scripts used to calculate shape parameters are freely available (Zheng and Hryciw, 2015). Source codes of Rgrains is freely available (Yamada, 2024). Also, Rgrains can be run without a MATLAB license. We mentioned these points in the manuscript.

Line 171: The statement may be accurate, but please provide a reference or example to support it.
[AR] Line 208: We added a reference that mentions the importance of contrast between target objects and the background when segmenting images.

Line 192: Authors mention converting pixels to millimeter. Was any correction applied for pixelisation effects? If so, please describe the method used.
[AR] (the corresponding sentence is removed because the related figure is removed) We did not apply any correction. The image resolution is at least 320 pixel/cm, so we think pixelisation effects have minor effect on converting pixels to millimeter.

Definition of shape parameters and downstream evolution model
This subsection may better fit in the method section.
[AR] Line 136: Moved to the beginning of the method section.

Lines 196–209 describe the calculation of rate changes and histogram comparisons under the "Image A nalysis" subsection. However, these analyses are not part of image processing per se. It would be more appropriate to move this content to a new or existing section—such as "Definition of Shape Parameter s and Downstream Evolution Model".
[AR] Line 238: Moved to a new section "3.4 Downstream evolution and changing rates of shape param eters".

Line 196: The manuscript states that median values of R and IR were used. Please justify how the medi an represents the sampling population, especially given lateral sediment inputs from tributaries and hil lslopes.
[AR] Line 240: We have justified this point in the revised manuscript. There are two reasons why we u sed median. Firstly, some distributions of R and IR are clearly skewed, and thus median values are clos er to mode values than mean values (All distributions of R and IR are shown in figshare: Takahashi et a l., 2025, https://doi.org/10.6084/m9.figshare.28424138). Secondly, we have checked downstream cha nges in 16th and 84th pecentile values of R and IR and found that they are essentially similar to the cha nge in median R and IR. We added figures showing changes in 16th and 84th pecentile values to figsha re.

The exponential model used to estimate rates of change does not account for the three-stage evolution proposed in the manuscript. If the model is not designed to capture this pattern, please clarify its suita bility for your objectives.
[AR] Line 158: The exponential model is necessary to show changing rates of R and IR are clearly diffe rent between the upstream and downstream section. We explained these points in the section "3.1 Defi nition of shape parameters and downstream evolution model".

Results
The section begins more like a figure caption. It would be better to start with a paragraph summarizing key findings and referencing specific subplots.
[AR] Line 298: The second paragraph was a summary of the results, so we have removed the first para graph that reads like a figure caption.

Figures 2a and 2b appear to be key to the manuscript, as they form the basis for the proposed three-sta ge evolution. It is therefore recommended that these two plots be presented as a full-page figure to allo w readers to evaluate the data more easily. While the data points in Figure 2b are legible, those in Figur e 2a are overlapping, making it difficult to discern the trends for each grain size fraction.
[AR] Line 309, Fig. 3: We enlarged the figure and changed line properties to make it easier to distingui sh results of different sizes. Although it was not possible to completely avoid overlapping, the revised fi gure should be clearler than the previous version.

It is also recommended to use a distinct symbol for samples collected from the tributary, as their round ness values should ideally align with the trend observed for the main trunk stream based on transport d istance.

[AR] Line 309, Fig. 3: We have used a different symbol for tributary samples (triangles) and mainstream samples (circles). To improve the visibility, we changed a symbol for talus samples from triangles to squares. We also changed line styles of each size class.

There is no strict standard for using upstream or downstream distance, it is generally preferable to use downstream distance (downstream transport distance), setting 0 km at the channel head, and to maintain consistency by referring to this convention throughout the text when describing key observations.
[AR] We set 0 km at the channel head and used downstream transport distance throughout the manuscript.

Figure descriptions should follow a logical sequence. For example, Figure 2a and 2b are discussed in line 231, but subplots 2c and 2d are skipped entirely. Likewise, Figure 3 has four subplots, none of which are clearly referenced in the text.
[AR] We removed figure 3 so subplots 2c and 2d appear sequentially.

In Figure 3, the individual parameters used to compute the roundness metric 'R' are shown. If there is a specific reason for plotting them separately, it should be explained in the methods section where these metrics are introduced.
[AR] We originally included figure 3 because we were not aware of studies that explicitly showed changes in individual parameters used to calculate Wadell's roundness. But we admit inferences drown from figure 3 are probably obvious, so we have removed figure 3.

Table 1 and 2 present rates of change derived from Equations 2 and 3. Please explain how these rates—calculated from a single exponential function—support the concept of a three-stage evolution.
[AR] Line 327 (Table 2), 372 (Table 3): We used the two tables to show the changing rates of shape parameters were different between the upstream and downstream section. After reconsidering the uncertainties, we found R and IR values did not change based on high p-values and large uncertainties relative to changes in shape parameters between adjacent sites. We modified the manuscript accordingly. We also found a minor error in the calculation of regression coefficients, so we corrected the coefficient values in the tables.

Line 282: Less rounded particles from tributaries are expected due to shorter transport distances. Consider using percentile distributions to better evaluate sediment sources within the mixed sampling population.
[AR] Line 309, Fig. 3, figshare (Takahashi et al., 2025): We have checked the difference in R and IR between the mainstream and tributaries and found it did not depend on the choice of percentile values (16th, 50th, and 84th). Also, downstream changes in R and IR are similar among the 16th, 50th, and 84th values of R and IR. Thus, we have used only median values in the revised manuscript.

Figure captions should be self-contained and informative. For example, the caption "(c, f) Comparison between the crushed and non-crushed 2–4 mm talus samples" implies no difference between panels c and f. Also, figure subplots and captions should follow sequential order. For instance, Figure 7 starts with subplots a and d instead of a and b.
[AR] Line 417, Fig. 8: Changed the panel labeling and the caption.

Line 348: A hypothesis is introduced in the results section, but hypotheses and statistical tests should be defined earlier—in the introduction or methods section.
[AR] Line 289: We moved the corresponding part to the methods section.

Discussion

Lines 364–368: While the observed rates of change may align with previous field observations, the proposed three-stage evolution is inconsistent with the theoretical and experimental models cited. Most prior work supports a two-stage evolution described by a single (often exponential) function, e.g., Krumbein (1941), Domokos et al. (2014), Miller et al. (2014), Roussillon et al. (2009), Pokhrel et al. (2024).

[AR] Line 435-444: In the initial version of the manuscript, we wrote the decrease in the changing rates with transport distance has been reported by previous studies. For the marginal increase in shape parameters, we wrote it was not consistent with previous studies. Thus, we did not write the three-stage evolution was consistent with findings of the previous studies. However, we understand that the original statement was misleading, so we have removed "three-stage evolution" from the paragraph and other parts of the manuscript. Also, we clarified the difference between the finding of previous studies and ours.

The observation that coarser basaltic fractions are more rounded than finer ones is interesting and author mentions can be explained by weathering behavior in igneous rocks. However, this may not apply universally—for instance, granitic sediments often show more rounded pebble size particles compared to cobble size. Consider discussing whether this pattern is rock-specific or grain-size dependent.

[AR] Line 592: Thanks for the suggestion. We added a sentence stating that the same resutls may not be obtained for other sites or rock types. Indeed, our results showed that roundness of shale particles did not change with particle size. So, we noted that the tendency of larger particles having greater roundness does not always apply to other sites or rock types. The dependency of shape parameters on particle size probably changes with rock properties and flow characteristics (e.g., magnitude and frequency of flood). Although there should be more factors involved, we were not able to discuss whether the size dependency of shape parameters was rock-specific or grain-size dependent because we failed to find studies that considered this issue in detail.

Conclusion

The current data and analyses do not provide sufficient scientific support to robustly conclude a three-stage evolution of particle shape in the headwater region.

[AR] Line 686: Based on your comments, we have reconsidered the results and have found that it is too early to generalize the three-stage evolution. Also, the proposed three-stage evolution applies only to changes in the average character of bulk sediment, while the existing two-stage evolution applies to shape evolution of both single particle and bulk sediment. So, to avoid confusion, we removed "three-stage evolution" from the manuscript.

---

## Author Response (AR2)

Line 259 of the track changes manuscript: percentiles should be 16th and 84th here, rather than 14th and 86th?

[AR] We corrected to 16th and 84th percentiles.

[AR] We added the following sentence to Acknowledgements:

We thank András A. Sipos and an anonymous reviewer for very helpful reviews and associate editor Fiona Clubb for handling the manuscript.